# The *Drosophila* circadian clock gene *cycle* controls the development of clock neurons

**Grace Biondi**[1], **Gina McCormick**[1], **Maria P. Fernandez**[1,2]*

**1** Department of Neuroscience and Behavior, Barnard College, New York, New York, United States of America, **2** Department of Biology, Indiana University Bloomington, Bloomington, Indiana, United States of America

\* fernanm@iu.edu

**Data Availability Statement:** All the data are freely available, without restrictions, and it can be found here: https://github.com/graceb4/cyc-ms-raw-data.

**Funding:** This work was supported by a National Science Foundation (NSF IOS 2239994) to M.P.F.

## Abstract

Daily behavioral and physiological rhythms are controlled by the brain's circadian timekeeping system, a synchronized network of neurons that maintains endogenous molecular oscillations. These oscillations are based on transcriptional feedback loops of clock genes, which in *Drosophila* include the transcriptional activators *Clock (Clk)* and *cycle (cyc)*. While the mechanisms underlying this molecular clock are very well characterized, the roles that the core clock genes play in neuronal physiology and development are much less understood. The *Drosophila* timekeeping center is composed of ~150 clock neurons, among which the four small ventral lateral neurons (sLN$_v$s) are the most dominant pacemakers under constant conditions. Here, we show that downregulating the clock gene *cyc* specifically in the *Pdf*-expressing neurons leads to decreased fasciculation both in larval and adult brains. This effect is due to a developmental role of *cyc*, as both knocking down *cyc* or expressing a dominant negative form of *cyc* exclusively during development lead to defasciculation phenotypes in adult clock neurons. *Clk* downregulation also leads to developmental effects on sLNv morphology. Our results reveal a non-circadian role for *cyc*, shedding light on the additional functions of circadian clock genes in the development of the nervous system.

## Author summary

Daily behaviors and physiological processes are governed by the brain's circadian clock, a network of neurons that regulates internal rhythms through cyclic gene expression. In *Drosophila* (fruit flies), the circadian clock consists of approximately 150 neurons, with a small subset playing a key role in maintaining these rhythms. While the molecular mechanisms of clock genes, including *Clock* (Clk) and *cycle* (cyc), are well understood in terms of timekeeping, their roles in other aspects of neuronal physiology are less explored. In this study, we found that the *cyc* gene, beyond its function in maintaining circadian rhythms, also influences the development of key clock neurons. Specifically, reducing *cyc* expression in a group of clock neurons during development leads to defects in the morphology of those neurons in both larval and adult stages and affects the ability of adult flies to maintain behavioral rhythms in the absence of environmental cues. These findings

This grant provided the salary for G.B. The funders had no role in study design, data collection and analysis, decision to publish, or preparation of the manuscript.

**Competing interests:** The authors have declared that no competing interests exist.

reveal that clock genes like *cyc* have important roles in brain development, highlighting their broader significance beyond circadian regulation. This work provides new insights into how genetic factors involved in timekeeping also contribute to the formation of neural circuits, expanding our understanding of the diverse functions of circadian clock genes.

## Introduction

The proper wiring of neuronal circuits during development is essential for the neuronal control of behavior. Across animal species, sleep/wake cycle rhythms, as well as many other behavioral and physiological rhythms, are controlled by the circadian timekeeping system, a network of neurons that maintains endogenous molecular oscillations and rhythmic behavior with a ~24 hour period [1]. The proper functioning of this circadian network requires the formation of synaptic and peptidergic connections during development [2,3].

The *Drosophila* circadian clock neuron network comprises ~150 neurons and is the functional equivalent of the mammalian suprachiasmatic nuclei, which contain 20,000 neurons in mice [4–6]. All circadian clock neurons contain an intracellular molecular clock consisting of a transcriptional feedback loop of clock genes[7]. CLOCK (CLK) and CYCLE (CYC) are heterodimeric transcriptional activators that directly activate transcription of the *period* (*per*) and *timeless* (*tim*) genes. PER and TIM encode repressors that inhibit CLK-CYC function. Subsequently, PER and TIM are degraded, which enables the cycle to reinitiate every morning. CLK and CYC also interact with other genes in a secondary circadian loop by activating the genes *vrille* (*vri*), and *Par domain proteinε* (*Pdp1ε*) [8, 9]. *Clk* and *cyc* expression can be detected in almost all clock neurons even before some of these neurons show molecular oscillations [10], suggesting that these genes serve functions that precede the establishment of molecular rhythms.

*Drosophila* clock neurons are classified into multiple clusters with distinct patterns of gene expression, anatomy, physiology, and synaptic connectivity [5, 6, 11–16]. Among these clusters, the small ventral lateral neurons (sLN$_v$s) are considered the most dominant pacemakers since they are critical for maintaining behavioral rhythmicity under constant darkness and temperature (DD, or free-running) [17–20]. The sLN$_v$s release the neuropeptide Pigment Dispersing Factor (PDF) [21], a key output signal within the clock neuron network [22]. PDF accumulates rhythmically at the sLN$_v$ dorsal termini [23] and can be released from both the neurites and soma [24]. Loss of PDF severely reduces the amplitude of the endogenous circadian rhythm and shortens its free-running period in DD [21]. The large LN$_v$s also produce PDF but do not play a role in maintaining rhythms in DD [17].

The projections of the four sLN$_v$s form a bundle and remain fasciculated as they extend from the ventral to the dorsal brain during development. These four projections are usually difficult to distinguish from each other until they begin to defasciculate in the dorsal protocerebrum [25] and extend their dorsal arborizations toward the area where dorsal clusters of clock neurons are located [26]. In adult flies, the dorsal termini of the sLN$_v$ projections show rhythmic structural plasticity [27], which relies on daily and circadian rhythms in outgrowth and fasciculation [28–31]. Both *Clk* and *cyc* mutants have lower *Pdf* RNA levels, and the PDF peptide can barely be detected in the sLNv projections [23, 32].

*Cyc* is a homolog of the mammalian gene *Bmal1*, although CYC protein levels do not cycle, unlike BMAL1 and several other *Drosophila* circadian proteins [33]. There is growing evidence for non-circadian functions of BMAL1. First, its downregulation induces apoptosis and cell-

cycle arrest in Glioblastoma Stem Cells (GSC), and it was found to preferentially bind metabolic genes and associate with active chromatin regions in GSCs [34]. Second, brain knockdown of *Bmal1* using CRISPR/Cas9 made glioblastomas grow at faster rates than controls [35], and similar effects were observed in B16 melanoma cells. Moreover, *Bmal1*(-/-) mice exhibit defects in short- and long-term memory formation [36] and show reduced lifespan and multiple symptoms of premature aging [37]. Overall, results from studies in different animal models suggest that *Bmal1* plays a role in the development of various neurological disorders [38].

The *Drosophila* sLN$_v$s offer an excellent model for exploring the non-circadian roles of canonical clock genes such as *cycle*. To determine if the phenotypes previously observed for *cyc* mutants are specific to PDF expression or involved a broader, non-circadian effect in the development of PDF- expressing cells, we downregulated *cyc* specifically in the *Pdf*-expressing cells and observed pronounced defasciculation of the sLN$_v$ projections. Similar phenotypes were observed upon expression of a dominant negative form of *cyc*. Moreover, we found that *cyc* downregulation in *Pdf*+ cells during development is sufficient to prevent the fasciculation of the adult sLN$_v$s and results in the loss of behavioral rhythms in adult flies. Manipulations of *Clk* expression also affect sLNv morphology, although remarkably, the phenotypes of *Clk* and *cyc* manipulation differ. Our results show that *cyc* plays a role in the development of pacemaker neurons, which is likely independent of its role in the circadian molecular oscillator.

## Results

### *cyc* downregulation in circadian pacemaker neurons affects the formation of sLN$_v$ axon bundles

Mutations in both *Clk* and *cyc* severely reduce *pdf* RNA and neuropeptide levels [23, 32]. In *cyc* null mutants, sLN$_v$s projections are often undetectable in larval and adult brains stained with PDF antibodies [23, 39, 40], although around half of the brains show 'stunted' projections [40]. We observed that *cyc* null mutants (*cyc$^{01}$*) showed a substantial reduction in PDF levels at ZT2, consistent with previous studies, but we also noticed the presence of thin, misrouted sLN$_v$ projections in *cyc$^{01}$* flies at higher magnification and intensity (S1A Fig). Upon close observation, PDF could often be detected in the sLN$_v$ projections. However, these projections did not form the stereotypical bundle observed in control brains when extending from the anterior medulla toward the dorsal area of the brain.

Because highly defasciculated projections might contribute to the weaker PDF levels observed in *cyc$^{01}$* mutants, we examined the structure of the sLN$_v$ projections using a *Pdf*-RFP transgene, in which a cytosolic Red Fluorescent Protein (RFP) is controlled by the *Pdf* regulatory sequence [41]. Flies were raised at 28°C throughout development, and experiments were conducted at 28°C in 6–8-day old flies (Fig 1A). In control brains, the projections from the four sLN$_v$s remain fasciculated, forming a bundle until reaching the superior medial protocerebrum (SMP). In contrast, the sLN$_v$s of *cyc$^{01}$* mutants often began to defasciculate in the ventral brain, near their cell bodies (Fig 1B). Some sLN$_v$ projections were severely misrouted and did not reach the dorsal brain, extending instead toward the midline or other brain regions. As a result, it was possible to distinguish individual projections from each sLN$_v$ even in the ventral brain in most *cyc$^{01}$* mutant brains. This is almost never observed in control brains until the sLNv projections reach the main branching point in the SMP. The morphological phenotypes of *cyc$^{01}$* flies are highly variable, and in some instances the projections are barely visible (S1B Fig).

To test whether the effect of *cyc* loss on the LN$_v$s is cell-autonomous, we next expressed a UAS-*cyc* dsRNA transgene (UAS-*cyc$^{RNAi}$*) using the *Pdf-Gal4* driver. We quantified the length

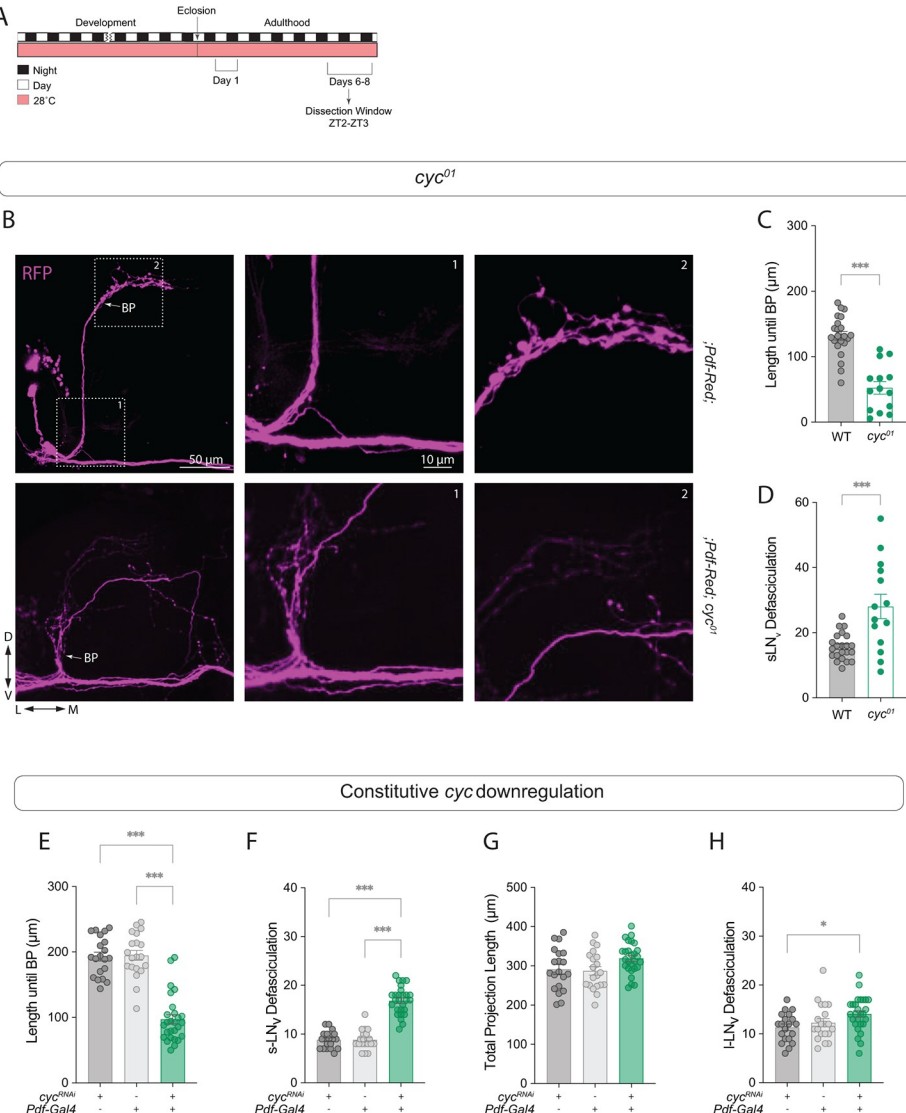

**Fig 1. *Cyc* downregulation in circadian pacemaker neurons prevents the formation of sLN$_v$s axon bundles.** The *cyc$^{01}$* mutant has disrupted sLN$_v$ morphology. (A) Representative timeline of the experiments in the figure. Flies were kept at 28˚C throughout development and experiments were performed within days 6–8 post-eclosion. (B) Representative confocal images of *Pdf-RFP* controls and*; Pdf-RFP;cyc$^{01}$* experimental flies stained with anti-RFP (magenta). The branching point (BP) of the dorsal projections is indicated. Scale bar = 25 μm. Boxes with dashed lines indicate the proximal (1) and distal (2) projections, corresponding to the labeled projection images in the center and right panels, respectively. An unpaired t-test was used to quantify the sLN$_v$ projection length until the branching point (BP) (C), and the total number of intersections of the sLN$_v$ ventral projections (D). Results from two independent experiments, with each dot representing one brain. For each genotype, the number of subjects (n) fall in the range: $13 \leq n \leq 22$. (E-H) Quantification of the LN$_v$ morphology phenotypes of experimental flies in which a *cyc$^{RNAi}$* transgene was driven by a; *Pdf-RFP,Pdf-Gal4;Tub-Gal80$^{ts}$* driver compared to the parental controls. The sLN$_v$ projection length until the branching point (BP) (E), the total number of intersections of the sLN$_v$ ventral projections (F), the total sLN$_v$ projection length (G), and the total number of intersections of the lLN$_v$ projections along the optic tract (OT) (H) are shown. Results from three independent experiments, with each dot representing one brain. For each genotype, the n falls in the range: $20 \leq n \leq 27$. For nonparametric data sets, statistical comparisons were done with Kruskal-Wallis tests followed by Dunn's multiple comparisons tests. For parametric data sets, statistical comparisons were done with one-way ANOVAs followed by Tukey post hoc tests. Differences that are not significant are not indicated. $^*p < 0.05$, $^{***}$ $p < 0.001$. Error bars indicate SEM.

of the projections, starting at the point where the projections of the sLN$_v$s intersect with those of the lLN$_v$s ("point of origin", POI, S1C Fig), until the first branching point ("branching point", BP). This branching point is where the sLN$_v$s ramify and extend their stereotypical arborizations in the dorsal protocerebrum in control brains, and these arborizations show daily, clock-controlled rhythms in their fasciculation and outgrowth [27]. Downregulating *cyc* in the *Pdf*-expressing cells significantly decreased the distance to the branching point (Fig 1E). Using a modified Scholl's analysis [42], we quantified the degree of branching in the ventral projections starting at the POI. We observed pronounced defasciculation in the sLN$_v$ projections in *Pdf* > *cyc*$^{RNAi}$ flies (Fig 1F). *cyc*$^{01}$ mutants also showed decreased distance to BP and sLN$_v$ fasciculation (Fig 1C and 1D). The total projection length in *Pdf* > *cyc*$^{RNAi}$ flies was not different from that of the controls (Fig 1G), and the defasciculation phenotype was not observed in the contralateral projections that extended from the lLN$_v$s (Fig 1H). Since the lLN$_v$s are born later in development during metamorphosis [25], this result suggests that *cyc* plays a role in neuronal development during an earlier developmental stage, when the sLN$_v$s begin to extend their projections toward the dorsal brain. These experiments were conducted at 28℃ to allow subsequent comparisons with adult-specific and development-specific downregulations of *cyc* using *Gal80*$^{ts}$. Similar results were obtained with flies raised at 25℃ (S1F–S1I Fig).

Expression of dominant negative forms of *Clk* and *cyc* is an effective strategy for preventing circadian molecular oscillations in specific groups of clock neurons [43–45]. In these dominant negative forms, the DNA binding ability is disrupted while the ability to heterodimerize is preserved [43]. Based on the phenotypes induced by *cyc* downregulation, we asked if expressing a dominant negative form of *cyc* in the sLN$_v$s also leads to aberrant projection morphology. We found that sLN$_v$s expressing Δ-*cyc* using *Pdf-Gal4* had a significantly shorter distance until the branching point (S2B and S2C Fig) and a greater degree of sLNv projection defasciculation (S2B and S2D Fig), similar to the effects observed in *Pdf* > *cyc*$^{RNAi}$ flies. The total projection length and the projections of the lLNvs were unaffected (S2E and S2F Fig).

## PER levels in *Pdf*+ neurons are reduced upon cell-specific *cyc* knockdown

CYC activates *per* transcription, and thus, PER levels in the brain are significantly reduced in *cyc* mutants [33]. To test whether the phenotypes of *cyc*$^{RNAi}$ expression in the *Pdf*-expressing neurons are consistent with what would be expected from *cyc* downregulation, we compared PER levels in the parental control (*Pdf-Gal4*, *Pdf-RFP*/+) with those in *Pdf* > *cyc*$^{RNAi}$ flies at the end of the night (ZT23), when PER nuclear levels are highest [46]. We found that nuclear PER levels in *Pdf* > *cyc*$^{RNAi}$ flies were significantly reduced in the sLN$_v$s (Fig 2B and 2C) and lLN$_v$s (Fig 2D and 2E). In contrast, PER levels were unaffected in the Dorsal Lateral Neurons (LN$_d$s) (Fig 2E and 2G). These results confirmed that, at least in a light-dark cycle (LD), *Pdf* > *cyc*$^{RNAi}$ flies have lower PER levels in *Pdf*$^+$ neurons.

*cyc* null mutant flies have pronounced behavioral phenotypes. Their activity is unimodal instead of bimodal during LD, and they are predominantly nocturnal [47]. Additionally, *cyc* mutants are largely arrhythmic in DD due to the key role of *cyc* in circadian molecular oscillations [33]. We conducted behavioral experiments to determine the extent to which downregulating *cyc* specifically in PDF-expressing neurons recapitulates the phenotype of the *cyc* mutant. We found that at 28˚C, the activity pattern of *Pdf* > *cyc*$^{RNAi}$ flies was still bimodal in LD (Fig 2H and 2I). However, the majority (~90%) of the experimental flies were arrhythmic in DD (Fig 2J). *Pdf* > Δ-*cyc* flies showed similar behavioral phenotypes (S2G and S2H Fig), consistent with what was reported for their free-running behavior at 25˚C [43].

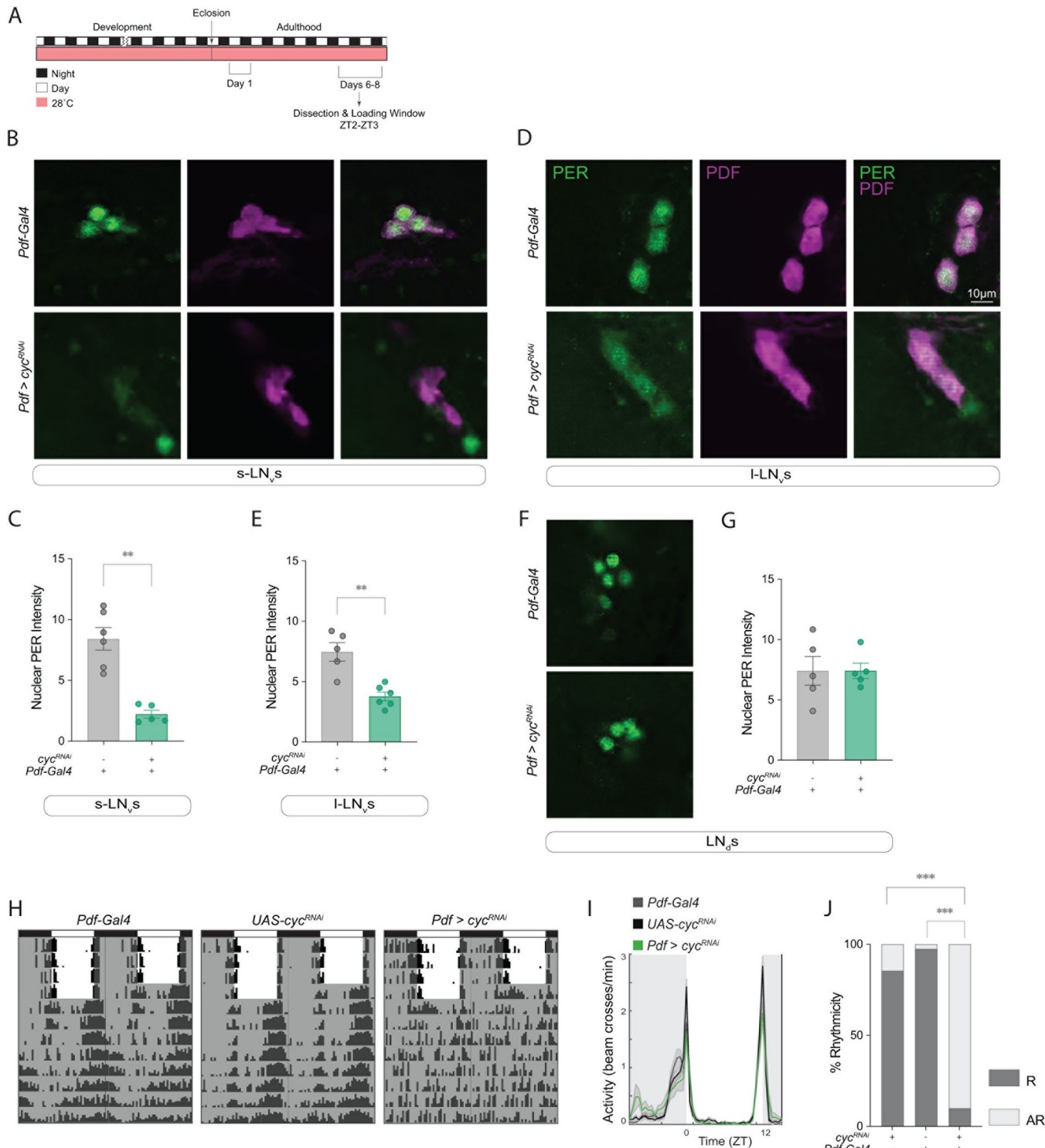

**Fig 2. Constitutive *cyc* downregulation in *Pdf*+ cells leads to a reduction in PER levels and arrhythmicity under free-running conditions.** (A) Representative timeline of the experiments in the figure. Flies were kept at 28˚C for their entire lifespan. Experiments were performed within days 6–8 post-eclosion. Dissections were performed at ZT2-3. (B,D,F) Representative confocal images of PER (green) and PDF (magenta) staining in the sLN$_v$s (B), lLN$_v$s (D), and LN$_d$s (F) of *Pdf > cyc$^{RNAi}$* experimental and *Pdf-Gal4* /+ control flies (n = 5–6 brains per clock neuron group). All lines also included a *Pdf-RFP* transgene. Scale bar = 10 μm. (C,E,G) Mann-Whitney tests were used to compare nuclear PER intensity levels in the sLN$_v$s (C), lLN$_v$s (E), and LN$_d$s (G) in flies of the indicated genotypes. Differences that are not significant are not indicated. ** p < 0.01. Error bars indicate SEM. (H) Representative actograms of flies of the indicated genotypes under 5 days of LD entrainment followed by 7 days of free-running (DD). To allow comparison with development-specific *cyc*

downregulation, flies in this experiment were raised at 28˚C for their entire lifespan and the experiment was conducted at 28˚C. (I) Population activity plots for flies during days 3–5 of the LD cycle at 28˚C. (J) Fisher's exact contingency tests were used to analyze the percentage of rhythmic flies of the indicated genotypes under DD (DD1-7). The driver line also included a *tub-Gal80$^{ts}$* transgene. Additional quantifications can be found in Table 1. R = Rhythmic and AR = arrhythmic. Differences that are not significant are not indicated. *** p < 0.001. Behavioral data corresponds to two independent behavior experiments. For each genotype: 40 ≤ n ≤ 48.

### *cyc* acts during development to shape neuronal morphology in adults

To knock down *cyc* specifically during development, we employed a temperature-sensitive Gal80 (Gal80$^{ts}$) variant with ubiquitous expression to conditionally inhibit Gal4-mediated expression of the RNAi [48]. This method enables the temporal regulation of UAS transgenes, as Gal80$^{ts}$ remains active at lower temperatures but becomes inactive at higher temperatures. We raised flies at 28˚C to allow *cyc* downregulation during development then transferred them to 18˚C immediately after eclosion (Fig 3A). After 1 week at 18˚C, the brains were dissected at ZT2 and stained with PDF and RFP antibodies (see methods section). As shown in Fig 3, downregulating *cyc* exclusively before eclosion resulted in abnormal morphology of the sLN$_v$ axonal projections in adult flies (Fig 3B). The phenotypes resembled those observed with constitutive downregulation, with a significantly shorter distance to the branching point (Fig 3B and 3C) and a greater degree of defasciculation compared to parental controls (Fig 3B–3D). No significant differences were found in the total projection length or the degree of defasciculation of the lLN$_v$s (Fig 3E and 3F).

A previous study showed that panneuronal rescue of *cyc* expression in a *cyc*$^{01}$ mutant exclusively during development was sufficient to partially rescue arrhythmicity in adult flies [40]. Therefore, we asked if downregulating *cyc* in the *Pdf*+ cells specifically during development would lead to behavioral phenotypes similar to those seen in the *cyc* null mutants. We found that under free-running conditions at 18˚C, most (~78%) of the *Pdf > cyc$^{RNAi}$* flies were arrhythmic (Fig 3G–3I). An analysis of PER subcellular localization in DD2 revealed clear cycling with nuclear PER higher at CT2 (Fig 3J). These results indicate that developmental downregulation of *cyc* specifically in the *Pdf*+ cells is sufficient to prevent behavioral rhythms in adults.

To determine whether adult-specific *cyc* knockdown in the *Pdf*-expressing cells would also lead to morphological phenotypes, we raised flies at 18˚C and switched them to 28˚C immediately after eclosion (S3A Fig, see the methods section). This manipulation did not result in morphological phenotypes either in terms of the length to the branching point or in the degree of sLN$_v$ defasciculation (S3B–S3F Fig). Under free-running 28˚C, the majority of the experimental flies were arrhythmic (S3G–S3H Fig), indicating that, as expected, *cyc* is required in adult clock neurons for proper circadian clock function.

### *Cyc* manipulations lead to aberrant sLN$_v$ projections in larval clock neurons

Next, we asked if *cyc* downregulation results in clock neuron morphology phenotypes during earlier developmental stages. The four larval sLNvs, which modulate the sensitivity of larvae to light and mediate a circadian rhythm in visual sensitivity [49], appear to be identical in their anatomy and synaptic connections [50]. We expressed the *cyc*$^{RNAi}$ transgene under the *Pdf-Gal4* driver and dissected third larval instar (L3) brains (Fig 4). In brains of experimental larvae the length to the branching point did not differ from that of the controls (Fig 4B and 4C), but the degree of dorsal termini branching was significantly higher (Fig 4B–4D). This quantification is similar to that previously described when quantifying the arborization of the dorsal projections sLN$_v$s in adults [27], where the concentric circles are centered at the main dorsal

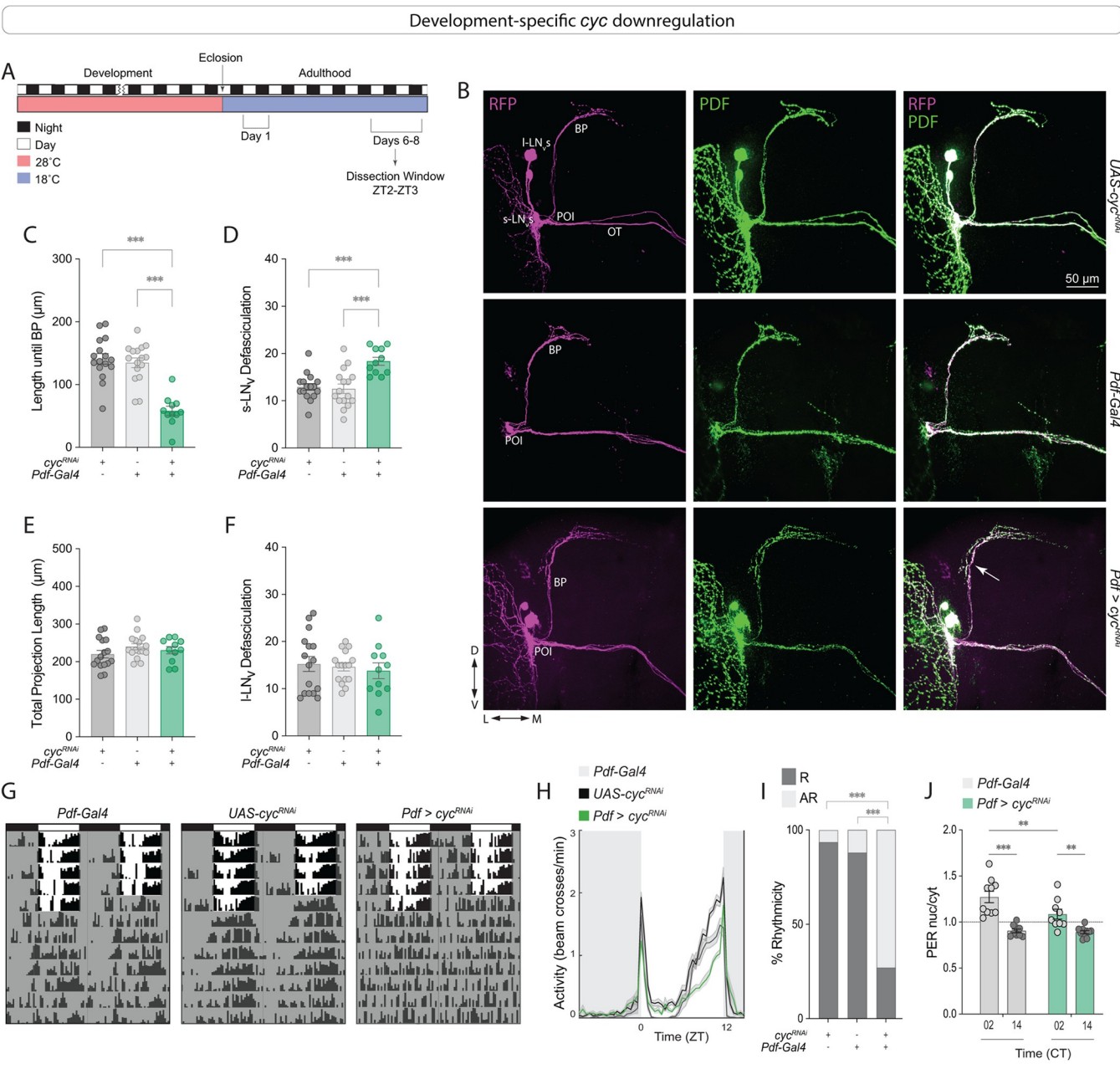

**Fig 3. Development-specific *cyc* downregulation in *Pdf*+ cells prevents sLNᵥ fasciculation.** (A) Representative timeline of the experiments in the figure. Flies were raised in LD at 28°C, and transferred to 18°C immediately after eclosion. Dissections were then performed in 6–8 day old adults at ZT2-3. (B) Representative confocal images of anti-PDF (green) and anti-RFP (magenta) staining of adult fly brains in which *cyc* was downregulated only during development. Each line also included a *Pdf-RFP* transgene. The white arrow indicates the increased defasciculation in the sLNv projections in experimental flies. Scale bar = 50 μm. (C-F) Quantification of the LNᵥ morphology phenotypes of flies of the indicated genotypes. The driver line also included a *tub-Gal80^{ts}* transgene. The sLNᵥ projection length until the branching point (BP) (C), the number of intersections of sLNᵥ ventral projections (D), the total sLNᵥ projection length (E), and the total number of intersections of the lLNᵥ projections along the optic tract (OT) (F) are shown for flies in which *cyc* was downregulated in *Pdf* + cells until eclosion. Two independent experiments were conducted. For each genotype: 11 ≤ n ≤ 16. One-way ANOVA tests were used to quantify the LNᵥ morphology. *** p < 0.001. Error bars indicate SEM. Each dot corresponds to one brain. (G-I). Behavioral phenotypes of development-specific *cyc* knockdown. Flies were raised in LD at 28°C, before being transferred to 18°C upon eclosion. Experiments were conducted at 18°C. (G) Representative actograms of flies of the indicated genotypes under free-running (see Table 1 for n and additional quantifications). (H) Population activity plots for flies during days 3–5 of the LD cycle at 18°C. (I) Percent rhythmicity for the indicated genotypes under DD. R = Rhythmic and AR = arrhythmic. Fisher's exact contingency tests were used to analyze the percentage of rhythmic flies under DD (DD1-7). *** p < 0.001. Error bars indicate SEM. The data correspond to three independent behavior experiments. For each genotype: 68 ≤ n ≤ 94. (J) Quantification of nuclear over cytoplasmic PER immunosignal within the sLNᵥs on day 2 of constant darkness at 18°C from brains of Gal4 controls or *cyc* RNAi-expressing flies. A two-way ANOVA was employed for statistical analysis. ** p < 0.01, *** p < 0.001. Error bars indicate SEM.

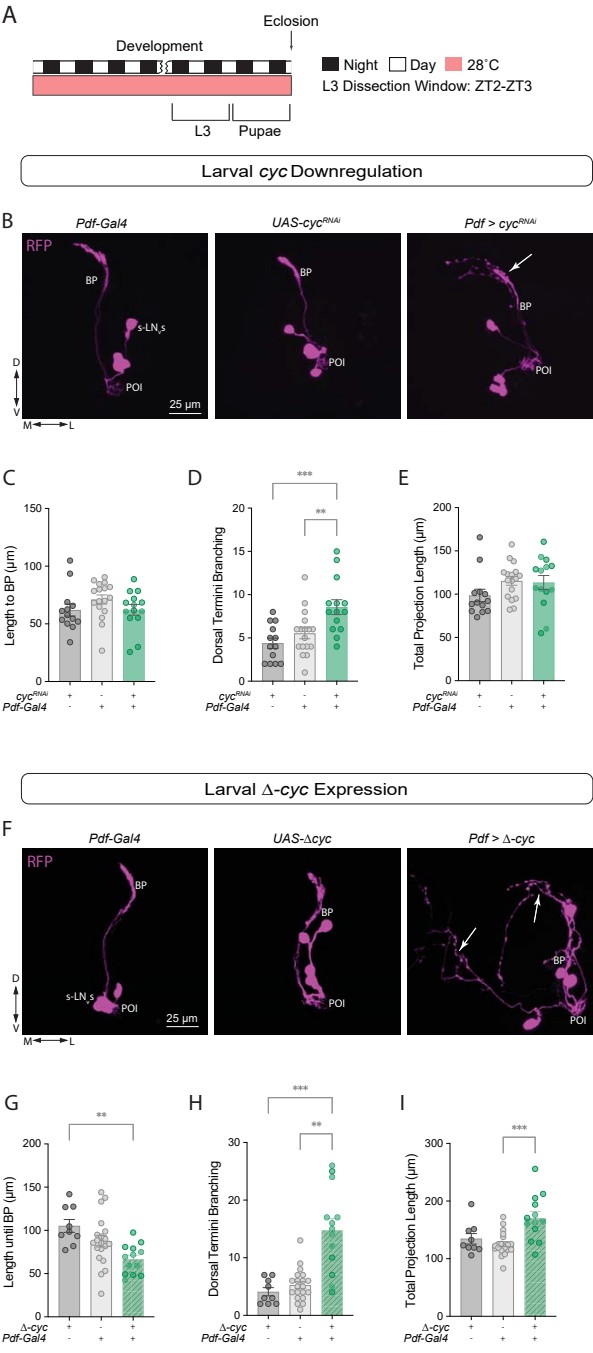

**Fig 4. *Cyc* manipulations lead to aberrant sLN<sub>v</sub> projections in larval clock neurons.** (A) Representative timeline of the experiments in the figure. Larvae were raised in LD at 28°C. Third instar larvae (L3) were dissected at ZT2-3. (B-E) Developmental effects of *cyc* knockdown in the sLN<sub>v</sub>s. (B) Representative confocal images of L3 larval brains stained with anti-RFP, labeling the sLN<sub>v</sub>s. (C-E) The projection length from the POI to the BP (C), the degree of sLN<sub>v</sub> dorsal termini branching (D), and the total projection length (E) were compared. For each genotype: $13 \leq n \leq 17$. (F-I) Developmental effects of expressing a dominant-negative form of *cyc*, *Δ-cyc*, in the larval sLN<sub>v</sub>s. (F) Representative confocal images of anti-RFP staining in the sLN<sub>v</sub>s of L3 larvae. A one-way ANOVA followed by a Tukey's Multiple Comparisons tests was used to compare the projection length from the POI to the BP (G). A Kruskal-Wallis tests followed by Dunn's multiple comparisons tests compared the nonparametric data sets: the degree of sLN<sub>v</sub> dorsal termini branching (H) and the total projection length (I). Each dot corresponds to one brain. For each genotype: $9 \leq n \leq 20$. ** $p < 0.01$, *** $p < 0.001$. Three independent experiments were conducted for each genetic manipulation and each line also included a *Pdf-RFP* transgene. The driver lines also included a *tub-Gal80<sup>ts</sup>* transgene. Error bars indicate SEM.

branching point (S1B Fig; see methods section). The total sLN$_v$ projection length was not affected by the genetic manipulation (Fig 4E).

Since the effects of *cyc* knockdown via RNAi and the expression of a *cyc* dominant negative form in adults were similar (Figs 1 and S2), we analyzed the morphology of the sLN$_v$ projections in L3 larvae upon *Δ-cyc* expression. In *Pdf > Δ-cyc* larval brains, the length to the branching point was significantly lower (Fig 4F and 4G) and the number of branches was significantly greater than that of controls (Fig 4H). The total projection length was not affected (Fig 4I). Taken together, these results suggest that *cyc* plays a role in the development of the larval sLN$_v$ neurons.

## *Clk* downregulation increases sLNv dorsal arborizations

*Clk* and *cyc* mutations produce similar effects on the expression pattern of PDF in adult brains [23]. CLK and CYC act as heterodimeric transcriptional activators, and the circadian phenotypes associated with mutations in these core circadian clock genes, both molecular and behavioral, are largely similar [43, 47, 51]. To determine if downregulating *Clk* in the sLN$_v$s leads to the same defasciculation of the sLN$_v$s observed with *cyc* manipulations, we performed similar experiments as those described above, in which we expressed *Clk*$^{RNAi}$ in *Pdf*+ neurons. We found that *Pdf > Clk*$^{RNAi}$ flies also showed neuronal morphology phenotypes (Fig 5).

In a previous study, *Clk* downregulation resulted in overfasciculation of the sLN$_v$ dorsal termini when stained with anti-PDF [52]. However, RFP labeling of the sLN$_v$ membrane indicated that these termini were actually more expanded than those of control flies, resulting in significantly higher dorsal termini branching (Figs 5B–5E and S1C). In *Pdf > Clk*$^{RNAi}$ flies, neither the distance to the branching point (Fig 5C) nor the degree of defasciculation differed from controls (Fig 5D). Neither the sLN$_v$ total projection length nor the lLN$_v$ projections were affected (S4C and S4D Fig). Only ~48% of the *Pdf > Clk*$^{RNAi}$ flies were rhythmic, and those that were rhythmic exhibited a lengthening of the free-running period (S4B Fig and Table 1). Nuclear PER levels in *Pdf > Clk*$^{RNAi}$ flies were significantly reduced in the LN$_v$s (S4H–S4J Fig).

Expression of *Δ-Clk* in the *Pdf*+ cells did not result in changes in the sLN$_v$ projection length until branching point (Fig 5F and 5G) or the total length of the projections (S4F Fig). However, the *Pdf > Δ-Clk* brains had increased defasciculation of the ventral projection (Fig 5H). The degree of dorsal termini branching in the *Pdf > Δ-Clk* flies was not significant (Fig 5I), not was the degreed of lLN$_v$ defasciculation (S4G Fig). Under DD at 28˚C, the majority of *Δ-Clk* expressing flies were arrhythmic (S4E Fig), consistent with what was reported at 25˚C [43].

We then examined L3 larval brains to determine if the observed phenotypes were already present at this developmental stage. While expression of *Clk*$^{RNAi}$ did not result in morphological phenotypes in larval LN$_v$s (S5 Fig), expression of *Δ-Clk* resulted in pronounced phenotypes (Fig 6A and 6B). We observed a significant increase in sLN$_v$ dorsal termini branching (Fig 6D) and total projection length (Fig 6E) in *Pdf > Δ-Clk* larvae. The length to the branching point for the experimental larvae was not significantly different from that of the control lines (Fig 6C). Expressing *Δ-Clk* led to more pronounced phenotypes in the larval stage than *Clk* downregulation, possibly due to incomplete knockdown.

In addition to the main feedback loop, CLK and CYC form a secondary loop by activating *vri* and *Pdp1ε* [8, 9], which repress and activate *Clk* expression, respectively. The low PDF peptide in the sLN$_v$s projections of *cyc*$^{01}$ mutants can be rescued by *vri* overexpression [39]. To determine if *vri* expression also affects sLN$_v$s morphology we expressed a line with a CRISPR/Cas9-based gRNA targeting the *vri* gene [53] under the control of the *Pdf*-Gal4 driver. We found that in *Pdf > Cas9 + vri-g* flies neither the distance until branching point in the s-LN$_v$s (Fig 7B and 7C) nor the degree of fasciculation of the s-LN$_v$s (Fig 7D) was different from

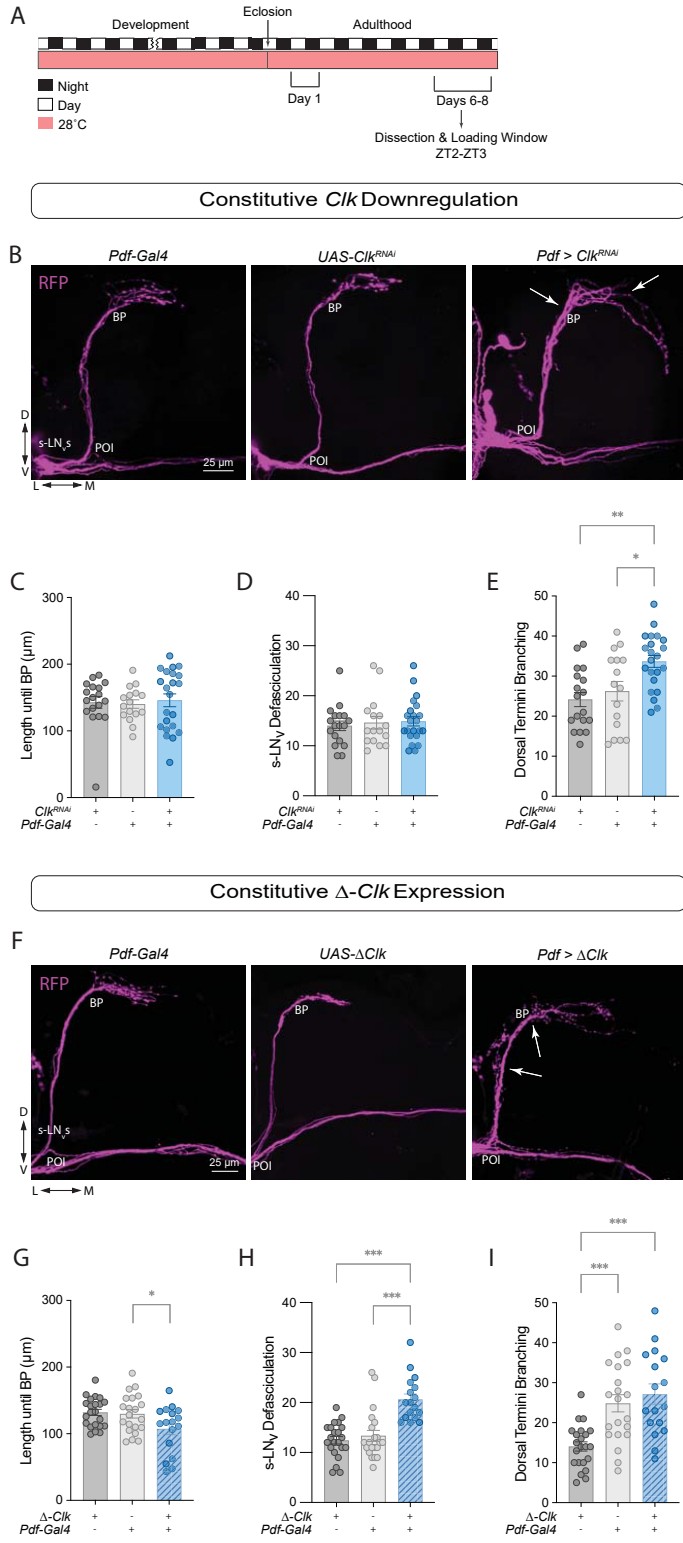

**Fig 5. *Clk* and *cyc* manipulations result in different morphology phenotypes in clock neurons.** (A) Representative timeline of the experiments in the figure. Flies were kept in LD conditions at 28˚C for their entire lifespan. Dissections were performed within Days 6–8 post-eclosion at ZT2-3. (B) Representative confocal images of anti-RFP staining in the sLN$_v$s adult brains of control (*Clk$^{RNAi}$* /+ and *Pdf-Gal4;tub-Gal80$^{ts}$*/+), and experimental (*Pdf > Clk$^{RNAi}$*) flies. White arrows indicate the BP (left) and extension of some of the sLN$_v$ dorsal projections (right) in the experimental

line. All lines employed also included a *Pdf-RFP* transgene. Scale bar = 25 μm. (C-E) Quantification of sLN$_v$ morphology using Kruskal-Wallis tests followed by Dunn's multiple comparisons tests for nonparametric datasets, compared the length until the branching point (C) and the total number of axonal crosses of the sLN$_v$s (D). For parametric data, ordinary one-way ANOVA tests followed by Tukey's Multiple Comparisons tests compared the total number of axonal crosses after the BP (E). Each dot corresponds to one brain. Two independent experiments were conducted. For each genotype: 16 ≤ n ≤ 22. * p < 0.05, *** p < 0.001. Error bars indicate SEM. (F) Representative confocal images of anti-RFP (magenta) staining in the sLN$_v$s adult brains of control (*UAS-ΔClk /+* and *Pdf-Gal4;tub-Gal80$^{ts}$/+*), and experimental (*Pdf > Δ-Clk*) flies. White arrows indicate the BP (top) and increased defasciculation along the sLN$_v$ projections (bottom). All lines employed also included a *Pdf-RFP* transgene. Scale bar = 25 μm. (G-I) Quantification of sLN$_v$ morphology phenotypes: length until the branching point (G), the total number of axonal crosses of the sLN$_v$s (H), and the total number of axonal crosses after the BP (I). For parametric data, ordinary one-way ANOVA tests followed by Tukey's Multiple Comparisons tests were employed. For nonparametric data, Kruskal-Wallis tests followed by Dunn's multiple comparisons tests were employed. See S1 Table for details about statistical analysis. Each dot corresponds to one brain. Two independent experiments were conducted. For each genotype: 17 ≤ n ≤ 22. ** p < 0.01, *** p < 0.001. Error bars indicate SEM.

controls. The total projection length was significantly higher than controls (Fig 7E), but in this case due to projections extending ventrally towards the optic tract (Fig 7B). In the majority of the brains, some s-LN$_v$ projections extended towards the ventral brain after reaching the SMP (Fig 7B) and in most cases contacted the l-LNv contralateral projections in the optic tract (Fig 7F and 7G), a phenotype that was never observed in control brains.

## Discussion

Our results reveal a role for the circadian clock gene *cyc* in establishing the proper cellular morphology of the key clock pacemaker neurons, the sLN$_v$s. Both constitutive *cyc* knockdown or expression of a dominant negative form of *cyc* in *Pdf*+ cells result in increased defasciculation of the sLN$_v$s. In addition, *Clk* downregulation and expression of a dominant negative form of *Clk* also result in sLN$_v$ morphology phenotypes, although some of those phenotypes appear to be distinct from those caused by *cyc* manipulations. Expressing the dominant-negative forms of either *Clk* or *cyc* has been used in previous studies as an effective way to prevent molecular oscillations in subsets of clock neurons. However, our results indicate that these genetic manipulations lead to additional morphological phenotypes beyond molecular time-keeping that are already detectable during the larval stages.

In addition to anatomical and functional classifications, clock neurons can be divided into early or late developmental groups depending on when circadian oscillations can be detected. In the early groups, which include the sLN$_v$s, *per* and *tim* expression rhythms can be detected at the first instar (L1) larval stage, whereas in the late groups, such rhythms cannot be detected until metamorphosis [25, 54]. However, *cyc* and *Clk* expression using GFP-*cyc* and GFP-*Clk* transgenes can be detected in almost all groups of clock neurons at early developmental stages, even days before *per* oscillations begin [10]. This suggests that *cyc* and *Clk* play additional roles in the development of clock neurons beyond their role in the molecular oscillator.

Both *cyc* and *Clk* modulate PDF expression in both larval and adult clock neurons. In *Clk*$^{jrk}$ mutants, neither PDF nor *Pdf* mRNA can be detected in most larval [32] or adult sLN$_v$s [23], and similar effects have been observed for the *cyc*$^{02}$ mutant [23]. However, around half of the *cyc*$^{01}$ brains stained with PDF exhibit 'stunted' sLNv projections which appear to lack their dorsal termini [40]. This study by Goda et al. also showed that panneuronal rescue of *cyc* expression throughout development is sufficient to restore PDF expression in the LN$_v$ dorsal projections of *cyc*$^{01}$ mutants [40]. Overexpression of *vri*, a clock gene that is downstream of CLK/CYC and acts as a repressor of CLK transcription [9, 32], causes a severe reduction in PDF levels in larval brains [32], and the low PDF levels in the sLN$_v$s of *cyc*$^{01}$ mutants can be

**Table 1. Summary of free running activity rhythms.** Related to Figs 2, 3, and 7 and S1–S4. Activity analysis of the above genotypes at 25˚C, 28˚C, or 18˚C. Light conditions for each experiment was 12:12 LD for 5 days followed by DD for at least 8 days. Depending on the experiment, flies were raised at 25˚C, 28˚C, or 18˚C. Flies raised a 25˚C were kept at that temperature throughout the behavior experiment. Flies raised at 18˚C were transferred to 28˚C upon eclosion and behavior experiments were conducted at 28˚C. Flies raised at 28˚C were either kept at 28˚C for behavior experiments (constitutive knockdown) or transferred to 18˚C upon eclosion (development-specific knockdown). ClockLab's $\chi$-square periodogram analysis was used to analyze rhythmicity, rhythmic power, and free-running period for each above genotypes. The % rhythmicity along with the number of rhythmic flies (nR), the period in hours with the SEM, and the rhythmic power with the SEM are indicated. Arrhythmic flies were not included in the analysis of period or power.

| Temperature pre-eclosion: 28˚C, Temperature post-eclosion: 28˚C | | | | |
|---|---|---|---|---|
| Genotype | Number of Flies (n) | % Rhythmicity (nR) | Period (h) ± SEM | Rhythmic Power ± SEM |
| ;Pdf-Red,Pdf-Gal4;Tub-Gal80$^{ts}$ | 40 | 97.50 (39) | 24.44 ± 0.06 | 109.80 ± 6.39 |
| ;UAS-cyc$^{RNAi\ 42563}$; | 48 | 85.42 (41) | 24.02 ± 0.08 | 85.74 ± 7.91 |
| ;Pdf-Red,Pdf-Gal4;Tub-Gal80$^{ts}$ >; UAS-cyc$^{RNAi\ 42563}$; | 40 | 10.00 (4) | 25.88 ± 3.03 | 24.32 ± 3.71 |
| ;UAS-Clk$^{RNAi\ 42566}$; | 26 | 92.31 (24) | 23.73 ± 0.10 | 97.66 ± 8.90 |
| ;Pdf-Red,Pdf-Gal4;Tub-Gal80$^{ts}$ >; UAS-Clk$^{RNAi\ 42566}$; | 21 | 52.38 (11) | 25.36 ± 0.22 | 36.74 ± 6.76 |
| ;Pdf-Red,Pdf-Gal4;Tub-Gal80$^{ts}$ | 24 | 100.00 (24) | 24.5 ± 0.07 | 101.40 ± 8.97 |
| ;UAS-cas9/CyO; UAS-Vrig/TM6b Tb | 16 | 87.50 (14) | 23.86 ± 0.11 | 124.6 ± 10.39 |
| ;Pdf-Red,Pdf-Gal4;Tub-Gal80$^{ts}$ >; UAS-cas9/CyO; UAS-Vrig/TM6b Tb | 18 | 38.89 (7) | 23.50 ± 0.15 | 29.02 ± 3.68 |
| ;Pdf-Red,Pdf-Gal4;Tub-Gal80$^{ts}$ | 31 | 83.87 (26) | 24.98 ± 0.07 | 56.76 ± 6.73 |
| ;UAS-Δcyc; | 24 | 95.83 (23) | 23.54 ± 0.08 | 72.79 ± 9.29 |
| ;Pdf-Red,Pdf-Gal4;Tub-Gal80$^{ts}$ >; UAS-Δcyc; | 32 | 3.13 (1) | 25.50 ± 0.00 | 13.66 ± 0.00 |
| ;UAS-ΔClk | 27 | 100.00 (27) | 23.52 ± 0.06 | 93.52 ± 8.71 |
| ;Pdf-Red,Pdf-Gal4;Tub-Gal80$^{ts}$ >; UAS-ΔClk | 28 | 3.57 (1) | 23.50 ± 0.00 | 17.18 ± 0.00 |

| Temperature pre-eclosion: 28˚C, Temperature post-eclosion: 18˚C | | | | |
|---|---|---|---|---|
| Genotype | Number of Flies (n) | % Rhythmicity (nR) | Period (h) ± SEM | Rhythmic Power ± SEM |
| ;Pdf-Red,Pdf-Gal4;Tub-Gal80$^{ts}$ | 68 | 88.24 (60) | 24.40 ± 0.17 | 87.80 ± 7.33 |
| ;UAS-cyc$^{RNAi\ 42563}$; | 94 | 93.62 (88) | 24.18 ± 0.04 | 81.06 ± 4.38 |
| ;Pdf-Red,Pdf-Gal4;Tub-Gal80$^{ts}$ >; UAS-cyc$^{RNAi\ 42563}$; | 70 | 27.14 (19) | 23.82 ± 0.81 | 21.42 ± 1.78 |

| Temperature pre-eclosion: 18˚C, Temperature post-eclosion: 28˚C | | | | |
|---|---|---|---|---|
| Genotype | Number of Flies (n) | % Rhythmicity (nR) | Period (h) ± SEM | Rhythmic Power ± SEM |
| ;Pdf-Red,Pdf-Gal4;Tub-Gal80$^{ts}$ | 30 | 100 (30) | 24.93 ± 0.04 | 127.8 ± 6.76 |
| ;UAS-cyc$^{RNAi\ 42563}$; | 25 | 96 (24) | 24.04 ± 0.07 | 150.1 ± 12.44 |
| ;Pdf-Red,Pdf-Gal4;Tub-Gal80$^{ts}$ >; UAS-cyc$^{RNAi\ 42563}$; | 31 | 38.71 (12) | 23.67 ± 0.61 | 24.27 ± 3.68 |

| Temperature pre-eclosion: 25˚C, Temperature post-eclosion: 25˚C | | | | |
|---|---|---|---|---|
| Genotype | Number of Flies (n) | % Rhythmicity (nR) | Period (h) ± SEM | Rhythmic Power ± SEM |
| ;Pdf-Red,Pdf-Gal4; | 29 | 89.66 (26) | 24.52 ± 0.08 | 152.2 ± 11.53 |
| ;UAS-cycRNAi 42563; | 31 | 83.87 (26) | 24.06 ± 0.08 | 95.44 ± 9.61 |
| ;Pdf-Red,Pdf-Gal4; >; UAS-cyc$^{RNAi\ 42563}$; | 31 | 12.90 (4) | 23.13 ± 0.24 | 21.55 ± 6.93 |

rescued by *vri* overexpression [39]. However, restoring PDF expression in the sLN$_v$s in flies lacking *vri* expression is not sufficient to rescue activity rhythms.

Unlike in *cyc* and *Clk* mutants [23], PDF can be detected in the sLN$_v$s projections in *per*[01] and *tim*[01] mutants, although it no longer shows rhythms in its accumulation in the dorsal termini [23]. In addition, structural plasticity rhythms in the sLN$_v$s are absent in both *per*[01] and *tim*[01] mutants [27]. Downregulation of *Clk* [52], expression of Δ-*cyc* [30], and overexpression of *vri* in *Pdf*-expressing cells [39] also result in impaired plasticity rhythms [55]. Although the

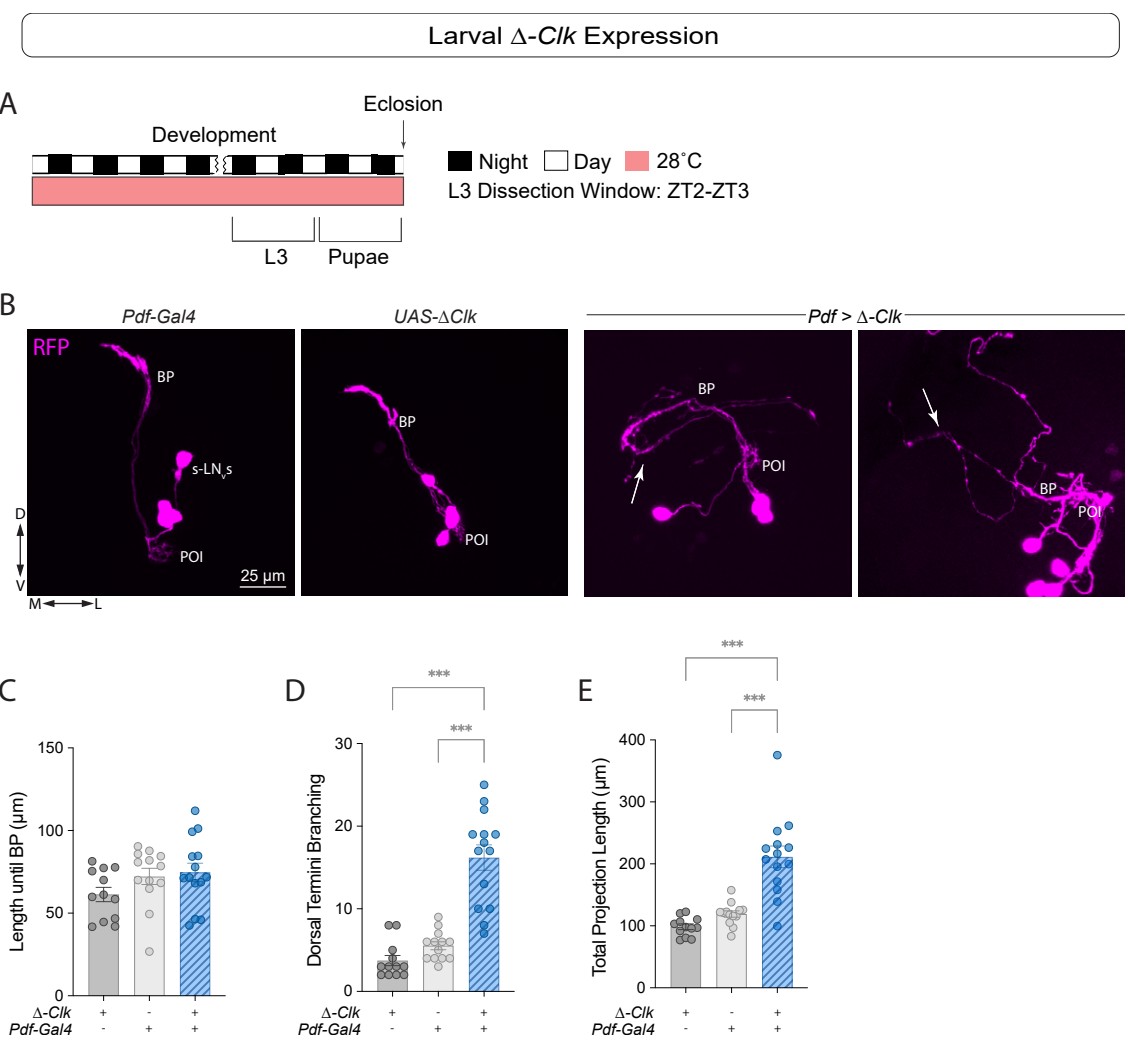

**Fig 6. Expressing *Δ-Clk* in the sLN_vs leads to axonal morphology phenotypes in L3 larvae.** (A) Representative timeline of the experiments in the figure. Larvae were raised in LD at 28°C. Third instar larvae (L3) were dissected at ZT2-3. (B) Representative confocal images of anti-RFP (magenta) staining in the sLN_vs when *Δ-Clk* was expressed in *Pdf+* neurons in L3 larvae. Each line also included a *Pdf-RFP* transgene, and the driver line also included a *tub-Gal80^{ts}* transgene. White arrows indicate misrouting of the sLN_v projections in the experimental line. Scale bar = 25 μm. For nonparametric data, Kruskal-Wallis tests followed by Dunn's multiple comparisons tests were used to compare the length to the BP (C). One way ANOVA tests were used to compare dorsal termini branching (D) and the total projection length (E). Two independent experiments were conducted. Each dot corresponds to one brain. For each genotype: 12 ≤ n ≤ 14. *** p < 0.001. Error bars indicate SEM.

anatomical phenotypes seen in these mutants are milder than those that observed when *cyc* and *Clk* are downregulated or when their dominant negative forms are expressed, the sLN_v projections of both *per* and *tim* null mutants also exhibit altered morphology [27].

Our results suggest that *Clk* and *cyc* manipulations produce different phenotypes, however, it is possible that this is partially due to a less effective knockdown of *Clk*. Behavioral experiments show that *cyc* knockdown in *Pdf+* neurons result in a larger fraction of arrhythmic flies than knockdown of *Clk* (Table 1). Use of RNAi often reduces gene expression but does not completely eliminate it, and may lead to off-target side effects. In addition, RNAi efficiency may vary over time. Expression of dominant negative alleles was used as an independent approach, but this method has limitations as well: over-expression levels for *cyc* and *Clk* may

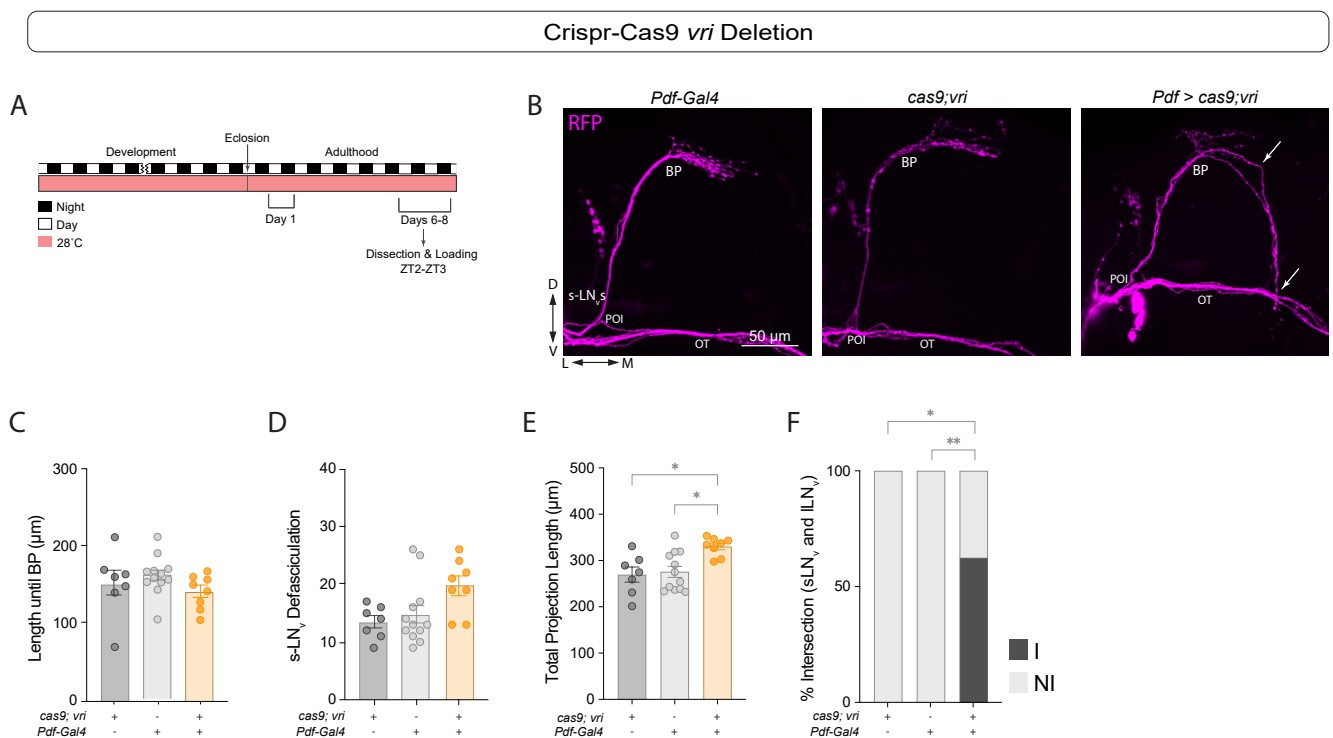

**Fig 7. *Vri* mutagenesis results in sLN$_v$ hyperextension.** (A) Representative timeline of the experiments in the figure. Flies were kept in LD conditions at 28˚C for their entire lifespan. Dissections were performed in 6–8 day old adults at ZT2-3. Behavioral experiments were run at constant 28˚C. (B) Representative confocal images of anti-RFP (magenta) staining in the sLN$_v$s adult brains of control (*cas9;vrig*\/+ and *Pdf-Gal4;tub-Gal80^ts*/+), and experimental (*Pdf > cas9; vrig*) flies. All lines employed also included a *Pdf-RFP* transgene. White arrows indicate the misrouting of the sLN$_v$ dorsal projections (top), and the intersection of the sLN$_v$ with the lLN$_v$s at the OT (bottom). Scale bar = 25 μm. (C-E) Kruskal-Wallis tests followed by Dunn's multiple comparisons tests were used to compare the length until the branching point (C), the total number of intersections of the sLN$_v$s ventral projections (D), and the longest path of the sLN$_v$ projections (without including misrouting) (E). (F) Fisher's exact contingency tests were used to analyze the percentage of brains where the sLN$_v$s intersected with the lLN$_v$s at the optic tract (I = Intersecting, N.I. = Not Intersecting). See Table 1 for additional quantifications. Each dot corresponds to one brain. Two independent experiments were conducted. For each genotype: 7 ≤ n ≤ 12. * $p < 0.05$, ** $p < 0.01$, *** $p < 0.001$. Error bars indicate SEM.

differ, and non-native molecular interactions may occur at high concentrations. However, differential effects of *cyc* and *Clk* mutations have been previously described: *cyc^01* and *Clk*^jrk mutants showed differences in their sleep consolidation during the day and in their ability to recover after sleep deprivation [47].

CYC/CLK may regulate neuronal fasciculation by modulating the expression of genes involved in cell migration or cytoskeletal dynamics. For example, increasing matrix metalloproteinases 1 (MMP1) expression reduces the complexity of the sLNv arborizations along the projections [31]. MMP1 promotes fasciculation in *Drosophila* motor neuron axons [56]. *Clk* has been shown to affect sLNv dorsal termini arborization through the activation of Mef2, which negative regulates Fas2 expression [52]. Our results from *Clk* downregulation show increased rather than decreased sLNv dorsal termini arborization. One important difference is that we used RFP to label the membrane rather than a PDF staining. As for *cyc*, among the sLNv morphology phenotypes reported in the literature, including those of other clock mutants, PDF/PDFR [57], and Rho GTPases [28], among others, the phenotype most similar to *cyc* downregulation is the downregulation of the *Medea (Med)*. Med is homolog of the human tumor-suppressor gene *DPC4* and is involved in the *decapentaplegic* (*dpp*) pathway [58], and its downregulation via RNAi in *Pdf*+ neurons results in decreased fasciculation along

the projections of the sLNvs [57]. In addition, similar to *cyc* manipulations, developmental specific downregulation of Med leads to morphology phenotypes in adult clock neurons [57].

Expression of *Clk* outside the clock network leads to the generation of ectopic clocks [59], but they require *cyc* expression. A study by Liu et al. showed that *Clk* stabilizes CYC both in cultured *Drosophila* Schneider 2 (S2) cells and *in vivo*: upon ectopic *Clk* expression, GFP-CYC can be detected in additional cells beyond the clock neuron network, suggesting that although with this reporter the CYC protein could be detected in the brain only in clock neurons, *cyc* mRNA is more broadly expressed [60]. In addition, *cyc* mRNA was not enriched in the LNvs compared to other *elav*-expressing neurons in the head [61]. Single cell RNA sequencing data revealed that *cyc* mRNA is present in non-clock neurons as well as in various tissues throughout the fly's body, with particularly high expression in the gut, ovaries, and testes [62]. In some instances, *cyc* mRNA expression levels are very high while *Clk* mRNA levels are low, such as in intestinal stem cells and the chordotonal organ [62]. The role of *cyc* mRNA expression in nonclock cells remains unknown. An interesting question for future studies is whether CLK and CYC act as an obligate heterodimer in their neurodevelopmental function and other possible non-circadian roles. In mammals, BMAL1 can dimerize with NPAS2 in addition to CLOCK [63], and a recent study in *Drosophila* detected co-binding of CYC and FOXO in the promoter region of *vrille* [64]. Single seq RNA sequencing in the sLNvs and other clock neurons, comparing the effects of *cyc* vs *Clk* downregulation, could help clarify the degree to which they function independently. Our results suggest that *Clk* and *cyc* are involved in shaping the morphology of clock neurons, and it is possible that they play similar roles in non-clock neurons as well.

## Materials and Methods

### Fly lines and rearing

Flies were raised on cornmeal-sucrose yeast media in a Percival Incubator under 12:12 LD at different temperature conditions. Depending on the experiment, flies were raised under either 18˚C, 25˚C, or 28˚C (indicated in the figure legends). The lines *UAS-cyc$^{RNAi}$* (BDSC #42563), *UAS-Clk$^{RNAi}$* (BDSC #42566), *w$^{1118}$* (BDSC #3605) and CS (BDSC #64349) were obtained from the Bloomington *Drosophila* Stock Center. The lines *Pdf-RFP,Pdf-Gal4;Tub-gal80$^{ts}$* and *w;Pdf-RFP;MKRS/TM6* were donated by Justin Blau (New York University). The *cyc$^{01}$*, UAS-*Δ-cyc*, and UAS-*Δ-Clk* stocks were donated by Paul Hardin (University of Texas).

### Immunohistochemistry

**LN$_v$ PDF levels and neuronal morphology.** Brains of 6–8-day-old adult males or L3 larvae were dissected between ZT2 and ZT3 in ice-cold Schneider's *Drosophila* Medium (S2) (Thermo Fisher, #21720024). They were fixed immediately after dissection in 2% Paraformaldehyde (PFA) in S2 for 30 minutes. Brains were then treated with blocking solution (5% goat serum in 0.3% PBS-Tx) for 1 hour at room temperature followed by incubation with primary antibodies at 4˚C for 24–48 hr. The primary antibodies used were 1:3000 mouse anti-PDF (Developmental Hybridoma Bank) and 1:1000 rabbit anti-RFP (Rockland, #600-401-379-RTU). After incubation, the brains were rinsed 6 times in 0.3% PBS + Triton X-100 (PBT), after which they were incubated with Alexa-fluor conjugated secondary antibodies for 24-hr at 4˚C. The secondary antibodies used were 1:3000 Alexa-488 (Thermo Fisher, #A11029) and 1:1000 Alexa-568 (Thermo Fisher, #A11036). The brain samples were further washed 6 times with 0.3% PBT, cleaned and mounted on a clean glass slide in Vectashield (Vector Laboratories, #H-1000-10) mounting media. A list of reagents can be found on Table 2.

**Table 2. List of reagents.**

| REAGENT or RESOURCE | SOURCE | IDENTIFIER |
|---|---|---|
| **Experimental Models: Organisms/Strains** | | |
| *w;Pdf-RFP,Pdf-Gal4;Tub-gal80<sup>ts</sup>* | J. Blau, NYU | |
| *w;Pdf-RFP;MKRS/TM6* | J. Blau, NYU | |
| *;UAS-ΔClk #1* | J. Blau, NYU | |
| *W;UAS-cas9/CyO;UAS-Vrig;TM6b Tb* | M.Rosbash, Brandeis | |
| *w;;UAS-cyc$^{RNAi42563}$* | Bloomington *Drosophila* Stock Center | BDSC 42563 |
| *w;;UAS-Clk$^{RNAi\ 42566}$* | Bloomington *Drosophila* Stock Center | BDSC 42566 |
| *w$^{1118}$;+;+* | Bloomington *Drosophila* Stock Center | BDSC 3605 |
| Canton-S | Bloomington *Drosophila* Stock Center | BDSC 64349 |
| *cyc$^{01}$* | P. Hardin, University of Texas | BDSC 80929 |
| *;UAS-Δcyc;* | P. Hardin, University of Texas | |
| *;UAS-ΔClk* | P. Hardin, University of Texas; Bloomington *Drosophila* Stock Center | BDSC 3618 |
| **Antibodies** | | |
| Rabbi anti-RFP (1:1000) | Rockland | #600-401-379-RTU |
| Mouse anti-PDF (1:3000) | Developmental Hybridoma Bank | |
| Rat anti-PER (1:500) | O. Shafer (ASRC CUNY) | |
| Anti-rabbit Alexa-568 (1:1000) | Thermo Fisher | A11036 |
| Donkey anti-rat Alexa-488 (1:500) | Thermo Fisher | A21208 |
| Anti-mouse Alexa-488 (1:3000) | Thermo Fisher | A11029 |
| **Software** | | |
| Fiji | http://fiji.sc | RRID: SCR_002285 |
| MATLAB R2022b | MathWorks, Natick | RRID: SCR_001622 |
| GraphPad Prism 9.0 | GraphPad Software | RRID: SCR_002798 |
| DAM FileScan | Trikinetics | |
| ClockLab | Actimetrics | RRID:SCR_014309 |
| **Chemicals, Peptides, and Recombinant Proteins** | | |
| Vectashield Mounting Medium | Vector Laboratories | #H-1000-10 |
| Premix PBS Buffer (10x) | Sigma-Aldrich | Cat# 11666789001 |
| 2% Paraformaldehyde (PFA) | Sigma-Aldrich | 47608-250ML-F |
| Triton- X-100 | Bio Basic | CAS#9002-93-1 |
| Schneider's *Drosophila* Medium (S2) | Thermo Fisher | 21720024 |
| **Other** | | |
| DAM2 *Drosophila* Activity Monitors | Trikinetics | |
| DAM Drosophila Environmental Monitors | Trikinetics | |

**PER Staining.** Brains of 6–8-day-old males were dissected one hour before lights-on (ZT23) in ice-cold Schneider's *Drosophila* Medium (S2) (Thermo Fisher, #21720024). Immediately after dissection, brains were fixed in 2% paraformaldehyde (PFA) for 30 minutes, stained and mounted as described above. The primary antibodies used were 1:1000 rabbit anti-RFP (Rockland, #600-401-379-RTU) and 1:500 rat anti-PER (donated by Orie Shafer). The secondary antibodies used were 1:1000 Alexa-568 (Thermo Fisher, #A11036) and 1:500 Alexa-488 (Thermo Fisher, #A21208).

For the analysis of PER subcellular localization (Fig 3J), flies were raised at 28C under LD and transferred to 18˚C immediately after eclosion. After 5 days under LD 18C, flies were transferred to constant darkness at 18˚C and brains were dissected on the second day of DD (DD2).

## Imaging, quantification, and statistical analysis

All images were acquired on an Olympus Fluoview 1000 laser-scanning confocal microscope using a 40x/1.10 NA FUMFL N objective (Olympus, Center Valley, PA) at the Advanced Science Research Center (ASRC-CUNY). For all the experiments, only one hemisphere per brain was imaged (the right hemisphere, unless it was damaged, in which case we imaged the left hemisphere).

<u>Quantification of adult $LN_v$ morphology ($sLN_v$ and $lLN_v$)</u>: We quantified 1) $sLN_v$ total projection length, 2) $sLN_v$ length from the point of origin (POI) until the branching point (BP) ('length until BP'), 3) the degree of defasciculation of the $sLN_v$ ventral projections, 4) the $sLN_v$ dorsal termini branching, 5) the degree of defasciculation of the l-LNv projections, and 6) intersections between $sLN_v$ projections and $lLN_v$ projections along the optic tract.

*1-Total projection length*: The total length of the dorsal projection was determined by a line drawn from the point of intersection (POI) between the $sLN_v$s and the optic tract until the end of the dorsal termini. If the projection length went past the midline of the brain, the length was measured up to the midline.

*2-Length until BP*: The partial length of the dorsal projections was determined by a line drawn from the POI until the BP of the $sLN_v$s at the dorsal termini. The projection length and partial projection length of the $sLN_v$s were quantified using Fiji in ImageJ.

*3- Defasciculation of the $sLN_v$ ventral projections ('$sLN_v$ Defasciculation')*: A modified Scholl's analysis [42], was used to analyze the degree of defasciculation of the ventral area of the $sLN_v$ projections, near the cell bodies. Six concentric circles, each 25 μm apart, were placed centered in the POI (S1C Fig). Each intersection between an individual ventral projection and any of the 6 circles was counted. A value of '10' denotes 10 total intersections between any of the projections and any of the circles.

*4-sLNv Dorsal termini branching*: A modified Scholl's analysis was used to analyze the degree of defasciculation of the dorsal termini of the $sLN_v$ projections. This method is similar to what was previously described to quantify sLNv dorsal termini [27]. In this study, 8 concentric circles, each 12.5 um apart, were centered in the BP (S1E Fig). Each intersection between an individual dorsal projection and any of the 8 circles was counted. A value of '10' denotes 10 total intersections between any of the dorsal projections and any of the circles.

*5-Defasciculation of the $lLN_v$ optic tract projections*. The degree of defasciculation of the lLNvs was determined using the same 6 concentric circles centered in the POI what were used to quantify defasciculation of the $sLN_v$ ventral projections (2) (S1C Fig). Each intersection between an individual lLNv projection and any of the 6 circles was counted.

*6- Intersections between $sLN_v$ projections and $lLN_v$ projections along the optic tract*. We quantified the percentage of brains in which at least one sLNv dorsal projection turned ventrally and extended towards the optic tract, contacting at least one lLNv projection. This phenotype was not observed in brains of control flies but was present in more than half of the brains of flies in which *vri* was knocked out (shown in Fig 7F).

**Quantification of larval $sLN_v$ morphology.** We quantified the total projection length of the $sLN_v$s, the axonal projection length until the branching point of the and the degree of branching in the $sLN_v$ dorsal projections (S1D Fig). The projection length, partial projection length, and area of the $sLN_v$s were quantified using Fiji. The projection length was measured by a line drawn from a determined first point of intersection (POI) of each of the $sLN_v$ cell bodies until the end of the dorsal termini. The partial length of the axonal projections was determined by a line drawn for the same point of intersection until the branching point (BP) of the $sLN_v$s at the dorsal termini. A modified Scholl's analysis was used to measure the branching of the $sLN_v$ projections. Six concentric circles were placed around the same

branching point used in the length measurements. The concentric circles were each 12.5 μm away from each other, so that the farthest circle was 75 μm away from the POI. The number of visible neurites of the sLN$_v$s that intersected with each circle were counted and summed, yielding the total number of intersecting neurons for the dorsal projections.

**Quantification of PER levels.** Single optical sections of either sLN$_v$s, lLN$_v$s or LN$_d$s were imaged using the same settings using a 40x/1.10 objective. PDF was used to identify the small and large LN$_v$s. The LN$_d$s were identified based on their localization, size, and morphology. PER levels were determined through normalization of nuclear staining within each cell to the background. The average value for each brain within a cluster was computed by averaging the values obtained from multiple cells within that cluster. Quantification was performed using images from 5–6 brains per each cluster at each timepoint. For the analysis of PER subcellular localization (Fig 3J), the ratio of nuclear vs cytoplasmic PER levels was determined for individual sLNvs and compared using a two-way ANOVA.

### Locomotor activity rhythm recording and analysis

DAM2 *Drosophila* Activity Monitors (TriKinetics, Waltham, MA) were used to record the locomotor activity rhythms of adult male flies aged three- to five-days, as previously described [65]. Flies were entrained to 12:12 LD cycles for at least five days, and then transferred to constant darkness (DD) for at least eight days at a constant temperature of 28°C, unless otherwise specified. Free-running activity rhythms were analyzed with ClockLab software from Actimetrics (Wilmette, IL). We employed ClockLab's χ-square periodogram function, which was integrated into ClockLab software, for the analysis of rhythmicity, rhythmic power, and free-running period in individual flies, using a confidence level of 0.01 [33]. For each of the tested genotypes, only significant periodicities falling within the 14 to 34-hour range were taken into consideration. In instances where an individual fly exhibited multiple periodicities with peaks surpassing the significance threshold, only the period with the highest amplitude was utilized when calculating the average periods presented in Table 1. ClockLab assigns each peak in the χ-square periodogram both a "Power" value and a "Significance" value. The "Rhythmic Power" for each designated rhythmic fly was determined by subtracting the "Significance" value from the "Power" value associated with the predominant peak. Flies that did not exhibit a periodicity peak above the threshold (10) were categorized as "arrhythmic," and their period and rhythmic power were not included in the analysis [65].

### Statistical analysis

Pearson's D'Agostino normality tests were performed for all the datasets. Depending on whether the data were normally distributed, statistical analyses were performed using either a one-way ANOVA with a Tukey's multiple comparisons test or a Kruskal-Wallis test with a Dunn's multiple comparisons test for 3 or more groups, or a t-test for comparisons between 2 groups. Fisher's exact contingency tests were run to analyze the percent rhythmicity for the indicated genotypes under DD.

### Supporting information

**S1 Fig. Quantification of LNvs defasciculation (ventral projections) and branching (dorsal projections).** (A-B) The *cyc$^{01}$* mutant has disrupted sLN$_v$ morphology. (A) Representative confocal images of anti-PDF staining in Canton-S control and *cyc$^{01}$* adult male brains. The sLN$_v$s and optic tract (OT) are indicated. Scale bar = 25 μm. The inserts on the right show the sLN$_v$ projections with the signal intensity adjusted for visibility in the *cyc$^{01}$* mutants. The top insert shows the distal (dorsal) area and the bottom insert shows the proximal (ventral) area of

the sLN$_v$ projections. Scale bar = 10 μm. (B) Representative images of eight brains of; *Pdf-RFP; cyc$^{01}$* experimental flies stained with anti-RFP (magenta). Flies were raised at 28˚C. Most of *cyc$^{01}$* mutant flies (~78.5%, 11 out of 14 brains) exhibit severe phenotypes in their sLNv morphology compared to the effects of *cyc* downregulation in *Pdf*+ neurons. (C-E) Representative confocal images of adult (C, E) and L3 larvae (D) control brains stained with anti-RFP (magenta). (C) To determine the degree of defasciculation of the sLN$_v$ ventral projections in adult brains, 6 concentric circles separated by is 25 μm were centered at the point of intersection (POI), where the projections of the sLN$_v$s and those of the lLN$_v$s intersect. The most distant circle does not reach the main branching point (BP) in control brains; therefore, the dorsal termini are not included. The number of intersections between either the sLNvs or the lLN$_v$s and each concentric circle were quantified. (D) Dorsal projection branching in the larval sLN$_v$s was measured by counting the number of intersections the sLN$_v$s had at each of the 6 concentric circles separated by 12.5 μm. (E) Adult sLN$_v$ dorsal projection branching was measured by counting the number of intersections the sLN$_v$s had at each of 8 concentric circles separated by 12.5 μm. This was adapted from a previous study [27] to capture the hyperextended projection phenotype of *Pdf > Δ-Clk* flies. (F-I) Quantification of the LN$_v$ morphology phenotypes of experimental flies in which a *cyc$^{RNAi}$* transgene was driven by a; *Pdf-RFP,Pdf-Gal4*; driver compared to the parental controls. Flies were raised at 25˚C. The sLN$_v$ projection length until the branching point (BP) (F), the total number of intersections of the sLN$_v$ ventral projections (G), the total sLN$_v$ projection length (H), and the total number of intersections of the lLN$_v$ projections along the optic tract (OT) (I) are shown. Graphs are representative of two independent experiments, with each dot representing one brain. For each genotype, n falls in the range: 18 ≤ n ≤ 21. (I-J). **p < 0.01, *** p < 0.001. Error bars indicate SEM. See S1 Table for additional quantifications.
(PDF)

**S2 Fig. *Δ-cyc* expression in *Pdf*+ cells prevents sLN$_v$s fasciculation.** (A) Representative timeline of the experiments in the figure. Flies were kept in LD conditions at 28˚C for their entire lifespan. Dissections were performed within days 6–8 post-eclosion at ZT2-3. (B) Representative brain confocal images of anti-PDF (green) and anti-RFP (magenta) staining in the sLN$_v$s of flies in which *Δ-cyc* was constitutively expressed in the *Pdf*+ cells using a *Pdf-Gal4;tub-Gal80$^{ts}$* driver. Each line also included a *Pdf-RFP* transgene. White arrows indicate branching of sLN$_v$ dorsal projections (left), and dorsal termini of the sLN$_v$ projections (right) for the experimental genotype. The images are representative of two independent experiments. Scale bar = 50 μm. LN$_v$ morphology was quantified by comparing the sLN$_v$ projection length until the branching point (C), the total number of intersections of the sLN$_v$ ventral projections (D), the full sLN$_v$ projection length (E), and the total number of lLN$_v$ intersections (F). One-way ANOVA was used to analyze normally distributed data (C, F). For nonparametric data sets, a Kruskal-Wallis tests followed by Dunn's multiple comparisons tests was used (D,E). * p < 0.05, ** p < 0.01, *** p < 0.001. Error bars indicate SEM. Each dot corresponds to one brain. For each genotype: 9 ≤ n ≤ 12. (G-H) Behavioral phenotypes of constitutive *Δ-cyc* expression. Experiments were conducted at 28˚C. (G) Population Activity (left) plots for flies during days 3–5 of the LD cycle at 28˚C (see Table 1 for additional quantifications). (H) Percent rhythmicity for the indicated genotypes under DD. R = Rhythmic and AR = arrhythmic. Fisher's exact contingency tests were used to analyze the percentage of rhythmic flies under DD (DD1-8). *** p < 0.001. Error bars indicate SEM. For each genotype: 24 ≤ n ≤ 32.
(PDF)

**S3 Fig. Adult-specific *cyc* knockdown does not affect sLNv neuronal morphology.** (A) Representative timeline of the experiments in the figure. Flies were raised in LD at 18˚C, and

transferred to 28˚C immediately after eclosion. Dissections were then performed within days 6–8 post-eclosion at ZT2-3. (B) Representative confocal images of anti-PDF (green) and anti-RFP (magenta) staining in the sLN$_v$s when *cyc* was downregulated exclusively after eclosion using a *Pdf-Gal4;tub-Gal80$^{ts}$* driver. The images are representative of two independent experiments. Scale bar = 50 μm. Each line also included a *Pdf-RFP* transgene. Kruskal-Wallis tests followed by Dunn's multiple comparisons tests were used to quantify the projection length until BP (C), the total number of intersections of the sLN$_v$ ventral projections (D), the full sLN$_v$ projection length (E), and the total number of lLN$_v$ intersections (F). * p < 0.05. Datasets are nonparametric (C-F). Each dot corresponds to one brain. For each genotype: 17 ≤ n ≤ 24. (G-H) Behavioral phenotypes of adult-specific *cyc* knockdown. Flies were raised in LD at 18˚C, before being transferred to 28˚C upon eclosion. Experiments were conducted at 28˚C. (G) Population Activity plots for flies during days 3–5 of the LD cycle at 18˚C (see Table 1 for additional quantifications). (H) Percent rhythmicity for the indicated genotypes under DD. Fisher's exact contingency tests were used to analyze the percentage of rhythmic flies under DD (DD1-8). *** p < 0.001. Error bars indicate SEM. For each genotype: 25 ≤ n ≤ 31. (PDF)

**S4 Fig. Expressing *Δ-Clk* in the LN$_v$s results in morphology and behavioral phenotypes.** (A) Representative timeline. Flies were raised in LD at 28˚C for their entire lifespan. Behavioral assays and dissections were performed within days 6–8 post-eclosion at ZT2-3. Experiments were conducted at 28˚C. (B) Fisher's tests were used to compare the percent of rhythmic flies of each indicated genotype (additional quantifications can be found in Table 1). ** P < 0.01, *** P < 0.001. For each genotype: 21 ≤ n ≤ 26. (C-D) Additional quantifications of effects of *Clk$^{RNAi}$* expression in the *Pdf+* cells in adult brains. Neither the sLN$_v$ total projection length (C) nor the lLN$_v$ projections (D) were affected. Datasets were quantified with ordinary one-way ANOVA tests followed by Tukey's Multiple Comparisons tests. For each genotype: 16 ≤ n ≤ 22. (E) Fisher's tests were used to compare the percent of rhythmic flies of each indicated genotype (additional quantifications shown in Table 1). For each genotype: 27 ≤ n ≤ 31. (F-G) Effects of *Δ-Clk* expression in the *Pdf+* cells in adult brains. The sLN$_v$ total projection length (F) and the lLN$_v$ projections (G) were quantified using one-way ANOVA tests followed by Tukey's Multiple Comparisons tests. Each dot corresponds to one brain. For each genotype: 17 ≤ n ≤ 22. (H-J) Mann-Whitney tests were used to compare nuclear PER intensity levels in the sLN$_v$s (H), lLN$_v$s (J), and LN$_d$s (J) in flies of the indicated genotypes. Flies were raised at constant 28˚C for their entire lifespan and dissections were performed at ZT2-3 * p < 0.05, ** P < 0.01, *** P < 0.001. Error bars indicate SEM. (PDF)

**S5 Fig. *Clk* downregulation in the larval sLN$_v$s did not result in morphology phenotypes.** (A) Representative timeline of the experiments in the figure. Larvae were raised in LD at 28˚C. Third instar larvae (L3) were dissected at ZT2-3. (B) Representative confocal images of anti-RFP (magenta) staining in the sLN$_v$s when *Clk$^{RNAi}$* was expressed in L3 larvae. Each line also contains a *Pdf-RFP* transgene. Scale bar = 25 μm. Kruskal-Wallis tests followed by Dunn's multiple comparisons tests were used to compare the projection length from the POI to the BP (C), the total number of axonal intersections (D), and the total projection length from the POI (E). * p < 0.05. Error bars indicate SEM. Each dot corresponds to one brain. For each genotype: 4 ≤ n ≤ 9. (PDF)

**S1 Table. Statistical Analysis for all Experiments.** Statistical analysis of each experiment, labelled with the corresponding figure, the genotypes used, and the comparisons between

genotypes. D'Agostino & Pearson Normality tests were employed to determine if datasets followed a normal distribution. For comparisons between two independent variables unpaired t-tests were used for parametric datasets, while Mann-Whitney tests were employed for non-parametric datasets. For comparisons between three independent variables ordinary one-way ANOVA tests followed by Tukey's Multiple Comparisons tests or Holm-Šídák's Multiple Comparisons tests were used for parametric datasets, while Kruskal-Wallis tests followed by Dunn's multiple comparisons tests were employed for nonparametric datasets. The p value for each test run is indicated, as is each tests corresponding significance. NS indicates results that are not significant. * p < 0.05, ** p < 0.01, *** p < 0.001.
(XLSX)

## Acknowledgments

We are very grateful to Justin Blau and Paul Hardin for their valuable feedback on various aspects of this project and for sharing fly lines with us. We are also grateful to M.Fernanda Ceriani, Amanda González-Segarra, Aishwarya Ramakrishnan Iyer, Orie Shafer, and Troy Shirangi for helpful comments on the manuscript, Annika Barber and Troy Shirangi for helpful discussions, Orie Shafer for the rat anti-PER antibody and Michael Rosbash for fly lines. The mouse anti-PDF antibody was obtained from the Developmental Studies Hybridoma Bank, created by the NICHD of the NIH and maintained at The University of Iowa, Department of Biology, Iowa City, IA 52242. Stocks obtained from the Bloomington *Drosophila* Stock Center (NIH P40OD018537) were used in this study.

## Author Contributions

**Conceptualization:** Maria P. Fernandez.

**Data curation:** Grace Biondi.

**Formal analysis:** Grace Biondi, Gina McCormick, Maria P. Fernandez.

**Funding acquisition:** Maria P. Fernandez.

**Investigation:** Grace Biondi, Gina McCormick, Maria P. Fernandez.

**Methodology:** Grace Biondi, Maria P. Fernandez.

**Project administration:** Maria P. Fernandez.

**Supervision:** Maria P. Fernandez.

**Visualization:** Grace Biondi, Gina McCormick.

**Writing – original draft:** Maria P. Fernandez.

**Writing – review & editing:** Grace Biondi, Maria P. Fernandez.

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
