## [Decision Letter · Decision Letter 0]

2 May 2024

Dear Dr Fernandez,

Thank you very much for submitting your Research Article entitled 'The Drosophila Circadian Clock Gene Cycle Controls the Development of Clock Neurons' to PLOS Genetics.

The manuscript was fully evaluated at the editorial level and by independent peer reviewers. The reviewers appreciated the attention to an important problem, but raised some substantial concerns about the current manuscript. Based on the reviews, we will not be able to accept this version of the manuscript, but we would be willing to review a much-revised version. We cannot, of course, promise publication at that time.

Should you decide to revise the manuscript for further consideration here, your revisions should address the specific points made by each reviewer. We will also require a detailed list of your responses to the review comments and a description of the changes you have made in the manuscript. In addition to responding to the specific comments from the reviewers, we feel that the manuscript would be strengthened if it included a more substantial discussion of its findings and their robustness.

If you decide to revise the manuscript for further consideration at PLOS Genetics, please aim to resubmit within the next 60 days, unless it will take extra time to address the concerns of the reviewers, in which case we would appreciate an expected resubmission date by email to plosgenetics@plos.org.

We are sorry that we cannot be more positive about your manuscript at this stage. Please do not hesitate to contact us if you have any concerns or questions.

Yours sincerely,

John Ewer

Academic Editor

PLOS Genetics

Gregory P. Copenhaver

Section Editor

PLOS Genetics

Reviewer's Responses to Questions

**Comments to the Authors:**

Reviewer #1: The manuscript examines LNv neuronal morphology in the cyc mutant background during both developmental and adult stages. The authors assert that CYC is essential for proper LNv neuronal morphology during development, in addition to its previously reported role in the molecular clock and claimed this is CYC’s non-circadian role. This point has been already brought up in previous reports (Ref# 23) but authors strengthen the point in this manuscript by scrutinizing the anatomical phenotype. However, further clarification and elaboration are needed, particularly regarding the anatomical phenotype of defective neurites in cyc mutant flies, identified as a fasciculation/de-fasciculation phenotype, which constitutes the primary focus of the manuscript (Main point #1). Additionally, it is somewhtat unsatisfactory that the manuscript remains descriptive of the phenomenon and lacks explanation of the underlying mechanism. It is suggested to include the mechanistic explanation of how cyc might affect neurite fasciculation in the discussion.

1. Figure 1: Given the critical role of the BP in the analysis, it is imperative to label the BP in every image, particularly in the mutant cases, similar to the labeling in Fig. 1C. Additionally, clarification is needed regarding the quantification of the degree of sLNv defasciculation. It remains unclear whether the authors refer to defasciculation of the axon bundle, dorsal termini, or both. The authors should elaborate on this aspect and provide a detailed explanation of how sLNv defasciculation was quantified.

2. Experimental Conditions: All experiments, except for Fig. 1A-C, were conducted at 28°C. Considering the influence of temperature on developmental speed and its contribution to morphological defects, it is necessary to justify the choice of temperature and explain why experiments were not conducted at the standard temperature of 25°C. Additionally, cyc01 phenotype is referenced to all other phenotypes, cyc01 experiment has to be done at the same temperature as other experiments, 28oC.

3. Pg5, line 129: The manuscript mentions that Cyc Ri flies resemble a minor version of the cyc01 phenotype without adequately describing the cyc01 phenotype categories. It is essential to distinguish between severe and minor phenotypes of cyc01 and provide the percentage distribution of these phenotypes.

4. Figure 3: Similar to Fig. 1, it is advisable to label the BP for other genotypes in Fig. 3. Clarification is needed regarding the white arrow. The authors should consider demonstrating the integrity of the molecular clock in the developmental stage KD condition to strengthen their argument regarding the molecular basis of arrhythmicity. Authors might show PER levels in the sLNV throughout the day.

5. Figure 5: The authors should provide evidence of Clk knockdown in ClkRNAi flies, such as showing PER or CLK expression levels. The statement regarding the differential effects of Clk and Cyc downregulation on neuronal morphology may need to be tempered, considering the similarity between Clkdelta and cyc downregulation phenotypes (cyc0, cycRi, cycdelta), which might be attributed to less efficient Clk knockdown. This apply to the statement in the abstract, the title of the result section in pg8 and the many places in the manuscript.

6. Figure 7F: The quantification of the distance from the OT appears somewhat arbitrary, and its significance is unclear. Instead, quantifying the intersection between dorsal projection and OT may provide a clearer explanation of the phenotype. Therefore It is suggested to denote intersection components in the axis title though.

7. Supplementary Figure 1: Similar to other figures, labeling the BP for every image is recommended. Additionally, the strong signal of concentric circles makes it difficult to visualize neurites, warranting consideration for improvement.

<minor>

1. Ensure proper citation for Fig. A in the text, and consider including a brief explanation of the experimental conditions.

2. Correct the typo on page 8, line 250, Pdf>delta-cyc larvae.

3. Pg. 13, line 370 if this reference 21 right?</minor>

Reviewer #2: This manuscript shed light on the role played by the key circadian transcription factor CYC in the development of circadian pacemaker neurons. It also explores how its partner CLK contribute to the development of these neurons. Surprisingly, CLK and CYC downregulation or dominant-negative mutants result in different phenotypes, suggesting that CLK and CYC have distinct functions, and would thus not work only as heterodimers in clock neurons.

The study is very clearly presented, the results are solid and they extend significantly our understanding of the developmental role of CLK and CYC, which had been previously noted but not studied in detail. It would be interesting to clarify if CLK and CYC’s developmental roles are completely independent of each other. It would thus be interesting to manipulate both genes simultaneously, and see if the phenotypes become a combination of both mutants, or whether one set of phenotype caused by downregulation of one protein is dependent on the activity of the other protein.

A few additional suggestions and questions:

1) Panel 3G: it seems the startle response occurs after the light-on transition. Is the LD shading on the actogram accurate?

2) Panel 3G and H. One panel 3G, experimental flies seem to be active randomly during the day, but on 3H they show clear evening anticipation, though reduced compared to control. Otherwise they show little activity during the day. Is the actogram on 3G really representative of the behavior of the experimental flies?

3) Flies with ablated LNvs usually show residual rhythmicity from non-PDF neurons. It seems that PDF-neuron specific downregulation shows a more severe phenotype, with no residual rhythmicity (panel 2H). Any speculation on why that is?

4) Line 310: I know what the authors meant to say, but this should be rephrased. CLK and CYC are expressed in many tissues, including glia, not just in clock neurons.

5) Line 311: Did single-cell RNA seq confirm the idea that cyc mRNAs are indeed present outside of the clock neurons? An alternative explanation would be that CLK works as a homodimer or as a heterodimer with an unknown partner to activate CYC expression and therefore the circadian transcriptional program in clock neurons, or in non-clock neurons when expressed ectopically. The present study indeed suggests CLK and CYC might be able to work independently of each other.

6) Line 166. I think the % AR flies is actually 90%, according to Table I. Line 189: 77%

Reviewer #3: This study addresses an interesting question, namely the dual role of the core Drosophila circadian transcription factors CLK and CYC in cell-specification and development versus molecular oscillator function.

Its conclusions are:

I) 'downregulating the clock gene cyc specifically in the Pdf-expressing neurons leads to decreased fasciculation both in larval and adult brains.'

II) 'This effect is due to a developmental role of cyc, as both knocking down cyc or expressing a dominant negative form of cyc exclusively during development lead to defasciculation phenotypes in adult clock neurons.'

III) 'Clk downregulation also leads to developmental effects on sLNv morphology, although cyc and Clk manipulations produce distinct phenotypes.'

IV)'Our results reveal a role for the circadian clock gene cyc in establishing the proper cellular morphology of the key clock pacemaker neurons, the sLNvs.'

The studys' experiments are generally presented to a high standard that gives confidence in the technical execution of the assays.

The main points that require a response are:

1) Mapping of a neurodevelopmental phenotype of cyc loss-of-function phenotype to the PDF-expressing neurons is not perfectly new. A neuronal morphology defect for Clk and cyc mutants was first recognized by Park et al., PNAS 2000 and then shown to arise developmentally by Goda et al., PLoS Genetics 2011 with the latter study also demonstrating that rescue of abnormal morphology mapped to the PDF neurons. A deficit of these prior studies was that imaging was conducted at lower resolution and with anti-PDF antibodies rather than with the use of reporters whose expression was less dependent on CLK/CYC activity per se. Nevertheless PDF signal was not completely absent in cyc01 mutant flies and the residual signal (reported for 25-56% of adult brains) was sufficient to conclude that the dorsal PDF+ s-LNv projections had abnormalities. The statement in lines 101-102 is, therefore, exaggerated. And the discussion of the prior literature should be refined without pre-empting the need for and interest in the presented new data, which shows the actual extent of the morphological phenotype.

2) Both transgenic RNAi and dominant-negative mis-expression approaches come with technical limitations that are not addressed in the discussion. I appreciate that knockdown efficiency is not easy to assess in adult s-LNv and that PER is used as a proxy. Nevertheless, using a single RNAi line for cyc and Clk makes interpretation of the results vulnerable to any (off-target) side-effects. Moreover, using the over-expression of dominant negative alleles, while providing an independent approach, is not without its caveats either depending on the achieved level of over-expression and the emergence of non-native molecular interactions at higher concentrations. While I am happy for the results to be published I would request that the limitations of the chosen approach are acknowledged.

3) I suspect that the authors agree that it is difficult to interpret the meaning of conclusion III. Given the technical limitations mentioned in point 2) above it may very well be that there is no reason to invoke novel partners or limited homodimerization for CLK and CYC in their neurodevelopmental role. The difference in phenotypes between Pdf>Δ-cyc and Pdf>Δ-Clk could reflect a difference in the native amounts of CLK versus CYC rather than a qualitative difference in the function of these transcription factors. Further, the behavioural phenotype of Pdf>ClkRNAi tubGal80ts at 28C in Table 1 is apparently milder than that for the comparable cycRNAi suggesting that knockdown efficiency and/or residual function may not have been the same. While the discussion conspicuously stays away from adressing this issues, I would prefer it if the question whether CLK and CYC act as an obligate heterodimer in their neurodevelopmental function was tackled, if only to formulate what future experiments might resolve this question.

4) I note that experimental conditions generally refer to constant 28C. This raises the question whether PDF s-LNv development in the presence of Pdf>cycRNAi or Pdf>ClkRNAi or Pdf>Δ-cyc or Pdf>Δ-Clk would be more normal in the absence of tubGal80ts at 25C. I am not asking for more experiments (unless the data is already available), but it should be acknowledged that development occurs more quickly at 28C and there may, therefore, be less room for error in patterning neuronal morphology at 28C than at a lower temperature, leaving open the possibility that neurodevelopmental phenotypes arise in a somewhat temperature-dependent manner regardless of the impact of temperature on the transgenic gene expression.

Table 1 is corrupted typographically and lacks statistical analyses. This needs to be remedied.

**Have all data underlying the figures and results presented in the manuscript been provided?**

Reviewer #1: Yes

Reviewer #2: Yes

Reviewer #3: **No: **Table 1 was corrupted and lacked statistical analyses

PLOS authors have the option to publish the peer review history of their article (what does this mean?). If published, this will include your full peer review and any attached files.

Reviewer #1: No

Reviewer #2: No

Reviewer #3: No

---

## [Decision Letter · Decision Letter 1]

23 Sep 2024

Dear Dr Fernandez,

Thank you very much for submitting your Research Article entitled 'The Drosophila Circadian Clock Gene Cycle Controls the Development of Clock Neurons' to PLOS Genetics.

The reviewers appreciated the revisions you made to the manuscript and are fully satisfied with the result. The "Minor revision" decision is rendered only so you consider the minor comment raised by Reviewer 2. Once this is addressed the manuscript will not go out for review again; the response to the reviewer's comment will be evaluated by the editor alone.

To resubmit, log into your Editorial Manager account and select the option 'Revise Submission' in the 'Submissions Needing Revision' folder.

Yours sincerely,

John Ewer

Academic Editor

PLOS Genetics

Gregory P. Copenhaver

Section Editor

PLOS Genetics

Reviewer's Responses to Questions

**Comments to the Authors:**

Reviewer #1: The authors have thoroughly addressed all of my previous concerns, and their revisions have significantly improved the overall quality of the manuscript. I find no further issues or concerns and this manuscript is now suitable for publication in PLOS Genetics.

Reviewer #2: The authors have addressed my comments. They should check again the actograms of figure 3G, as the LD cycle still seems slightly misaligned, by half hour I would guess.

Reviewer #3: The concerns that I raised have been addressed by the authors in their revised manuscript.

**Have all data underlying the figures and results presented in the manuscript been provided?**

Reviewer #1: Yes

Reviewer #2: Yes

Reviewer #3: Yes

PLOS authors have the option to publish the peer review history of their article (what does this mean?). If published, this will include your full peer review and any attached files.

Reviewer #1: No

Reviewer #2: No

Reviewer #3: **Yes: **Herman Wijnen

---

## [Editor Report · Decision Letter 2]

27 Sep 2024

Dear Dr Fernandez,

Many thanks for revising Fig 3G. Since this was the only (very minor) revision that was pending, we are pleased to inform you that your manuscript entitled "The Drosophila Circadian Clock Gene Cycle Controls the Development of Clock Neurons" has been editorially accepted for publication in PLOS Genetics. Congratulations!

Yours sincerely,

John Ewer

Academic Editor

PLOS Genetics

Gregory P. Copenhaver

Section Editor

PLOS Genetics

Comments from the reviewers (if applicable):

Many thanks for revising Figure 3G.

**Data Deposition**

http://datadryad.org/submit?journalID=pgenetics&manu=PGENETICS-D-24-00323R2

**Press Queries**

---

## [Editor Report · Acceptance letter]

11 Oct 2024

PGENETICS-D-24-00323R2 

The Drosophila Circadian Clock Gene Cycle Controls the Development of Clock Neurons 

Dear Dr Fernandez, 

We are pleased to inform you that your manuscript entitled "The Drosophila Circadian Clock Gene Cycle Controls the Development of Clock Neurons" has been formally accepted for publication in PLOS Genetics! Your manuscript is now with our production department and you will be notified of the publication date in due course.

With kind regards,

Jazmin Toth

PLOS Genetics

On behalf of:
